# Bridging the Knowledge-Prediction Gap in LLMs on Multiple-Choice Questions

Yoonah Park [* 1]   Haesung Pyun [* 1]   Yohan Jo [† 1]

## Abstract

While large language models (LLMs) perform strongly on diverse tasks, their trustworthiness is limited by erratic behavior that is unfaithful to their internal knowledge. In particular, LLMs often fail on multiple-choice questions (MCQs) even if they encode correct answers in their hidden representations, revealing a misalignment between internal knowledge and output behavior. We investigate and mitigate this *knowledge-prediction gap* on MCQs through a three-step analysis of hidden representations. First, we quantify the prevalence and magnitude of the gap across models and datasets. Second, we provide a geometric interpretation by identifying distinct *knowledge* and *prediction* subspaces in the residual stream. Third, we introduce **KAPPA**, a lightweight inference-time intervention that aligns the two subspaces within the residual stream to reduce the knowledge-prediction gap. Our results provide a geometric and interpretable explanation of the knowledge-prediction gap in LLMs. Furthermore, KAPPA effectively reduces the gap across diverse MCQ benchmarks and models, and generalizes to free-form settings.[1]

## 1. Introduction

While large language models (LLMs) exhibit remarkable performance across diverse tasks (Brown et al., 2020; Wei et al., 2022; Kojima et al., 2022), they often make unexpected errors such as hallucinations, systematic biases, and reasoning failures, raising concerns about their trustworthiness (Parrish et al., 2022; Shen et al., 2023; Bang et al., 2023; Ji et al., 2023; Dziri et al., 2023). In particular, on multiple-choice question (MCQ) benchmarks, a widely used

paradigm for evaluating LLMs, models frequently fail to predict correct answers, even on questions they appear capable of answering correctly. For example, a model may generate a correct answer in a free-form setting yet select an incorrect option when the same question is presented in an MCQ format (Figure 1a). Prior work has attributed such discrepancies to surface-level factors including option selection biases, superficial cues, or stylistic artifacts (Zheng et al., 2024; Balepur et al., 2025; Góral et al., 2025). However, a unified explanation that connects these failures to the model's internal representations remains unexplored.

Recent studies show that even when models answer incorrectly, their internal representations often linearly encode the correct answer (Liu et al., 2023; Orgad et al., 2025; Gekhman et al., 2025). Specifically, a simple linear classifier applied to the hidden states of an LLM frequently predicts the correct answer, even outperforming the model's own generation. This indicates that even when the correct answer is explicitly linearly encoded in the hidden states, models often fail to translate it into their final predictions. We term this phenomenon *the knowledge-prediction gap*: the discrepancy between the knowledge encoded in a model's internal representations and the predictions it ultimately generates.

Prior work has identified this gap primarily in limited settings such as truthfulness detection and simple arithmetic tasks (Marks & Tegmark, 2024; Azaria & Mitchell, 2023; Sun et al., 2025a), leaving several questions open: (i) whether this gap extends to diverse domains of MCQ tasks; (ii) what geometric structure underlies this gap in the model's hidden representations; and (iii) whether the gap can be mitigated at inference time without additional training. In this work, we address these questions by (i) establishing that the knowledge-prediction gap is widespread across MCQ benchmarks and model families, (ii) providing a geometric interpretation of this gap, and (iii) introducing an inference-time intervention that effectively mitigates this gap. We describe these contributions in turn below.

**First**, we establish that the knowledge-prediction gap is a widespread phenomenon in LLMs when solving MCQs. We train a knowledge probe—a linear $k$-class classifier that predicts the correct answer from the model's residual stream during question answering (Figure 1, Knowledge Probe). Using two complementary metrics, we measure the gap

---

[*]Equal contribution [†]Corresponding author [1]Graduate School of Data Science, Seoul National University. Correspondence to: Yoonah Park <wisdomsword21@snu.ac.kr>, Haesung Pyun <haesung.pyun@snu.ac.kr>, Yohan Jo <yohan.jo@snu.ac.kr>.

*Proceedings of the 43rd International Conference on Machine Learning*, Seoul, South Korea. PMLR 306, 2026. Copyright 2026 by the author(s).

[1]Repo: https://github.com/holi-lab/KAPPA

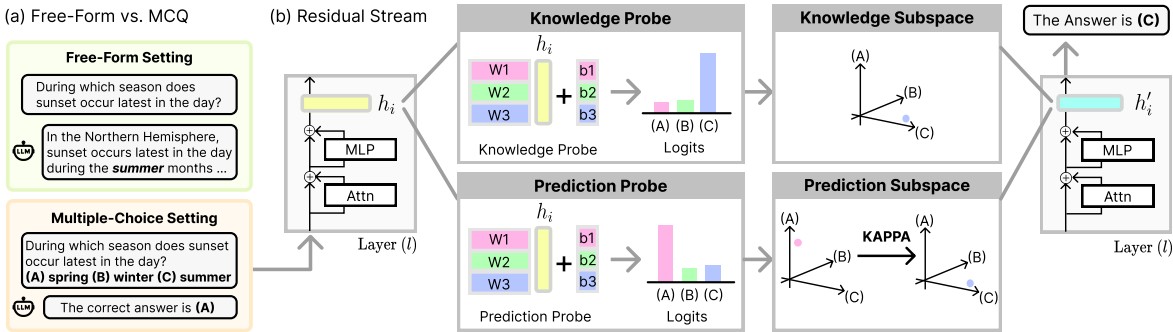

*Figure 1.* Given a hidden representation $h$ from the residual stream at layer $l$, we train two linear probes with disjoint objectives: a *knowledge probe* that predicts the ground-truth answer and a *prediction probe* that predicts the option decoded by the LLM. Although both probes operate on the same hidden state, their logits and the resulting coordinate representations differ. Interpreting probe weight vectors as defining low-dimensional subspaces, this discrepancy shows a misalignment between the coordinates of $h$ in the knowledge subspace and the prediction subspace. KAPPA modifies representations so that their coordinates in the prediction subspace match those in the knowledge subspace, thereby aligning model predictions with latent knowledge (described and evaluated in Section 5).

between this probe's predictions and the model's generated response, and find the gap to be pervasive, persisting across five representative models and a diverse set of MCQ benchmarks spanning reasoning, knowledge, truthfulness, and bias. The gap is most pronounced on reasoning, truthfulness, and bias benchmarks, while being less evident on knowledge-intensive benchmarks. Moreover, the gap remains consistent across model families and scales for truthfulness tasks, suggesting that it reflects a structural limitation rather than a model-specific artifact.

**Second**, we provide a geometric interpretation of the gap. We treat each probe's weight vectors as axes of a subspace, defining a *knowledge subspace* (from the knowledge probe) and a *prediction subspace* (from an analogous probe trained to predict the model's generated option; Figure 1, Prediction Subspace). The probe logits can then be interpreted as the hidden state's projection onto the corresponding subspace—its coordinate within that subspace. Ideally, the prediction coordinates, which indicate the option the model is likely to choose, should align with the knowledge coordinates, which indicate the option likely to be correct. However, on benchmarks with large knowledge-prediction gaps, we find substantial misalignment between these coordinates.

We further examine the gap by measuring representational alignment between the two subspaces. Using mean principal angles and centered kernel alignment (CKA; Kornblith et al., 2019), we show that the subspaces become geometrically distinct in deeper layers. This divergence strongly correlates with the measured knowledge-prediction gap across eight benchmarks, establishing that subspace misalignment systematically tracks the knowledge-prediction gap.

**Third**, we introduce an inference-time intervention to mitigate this knowledge-prediction gap. Our method, KAPPA (**K**nowledge-**A**ligned **P**rediction through **P**rojection-based **A**djustment), applies a closed-form affine transformation to

the residual stream that aligns each hidden state's coordinates in the prediction subspace with those in the knowledge subspace, while minimally perturbing other directions (Figure 1, KAPPA). Since the knowledge and prediction coordinates indicate the model's internal confidence over which option is likely correct and which option it is likely to output, respectively, this alignment steers the model's prediction toward the knowledge signal. As a result, KAPPA steers the model to act on its knowledge already encoded in its hidden states but not reflected in its prediction.

Our experiments show that KAPPA consistently reduces the knowledge-prediction gap and improves accuracy over existing inference-time approaches. These results suggest that many LLM failures stem not from missing knowledge, but from a misalignment between what the model internally encodes and what it ultimately predicts. Further analysis reveals that the identified knowledge and prediction subspaces partially transfer across tasks requiring similar skills, and even to free-form generation settings. Overall, our findings highlight representation-level alignment as a promising direction for eliciting more faithful predictions from LLMs.

Our contributions are threefold:

- We empirically reveal a **knowledge-prediction gap** in general MCQs by introducing quantitative metrics.
- We reveal the geometric structure underlying the gap: a misalignment between knowledge and prediction subspaces in the residual stream.
- We present KAPPA, an inference-time intervention that modifies hidden states to align a model's predictions with its internal knowledge.

## 2. Related Work

**The Knowledge-Prediction Gap.** Prior work shows that linear probes can extract correct answers from hidden

representations even when the model outputs are incorrect (Marks & Tegmark, 2024; Liu et al., 2023; Azaria & Mitchell, 2023; Su et al., 2024; Orgad et al., 2025; Gekhman et al., 2025). Several explanations have been proposed, including distractor-driven mechanisms that override correct knowledge (Tulchinskii et al., 2024; Wiegreffe et al., 2025; Yang et al., 2025) and miscalibration between early knowledge-encoding layers and later prediction-generating layers (Gottesman & Geva, 2024). However, most prior work examines this phenomenon in constrained settings such as truthfulness or arithmetic tasks (Liu et al., 2023; Sun et al., 2025a), and focuses primarily on identifying the gap rather than closing it.

**Inference-Time Intervention.** Recent work has explored modifying model behavior through representation-level interventions (Zou et al., 2025; Arditi et al., 2024; Rimsky et al., 2024; Hendel et al., 2023; Ilharco et al., 2023). Inference-time intervention methods typically train linear probes to identify meaningful directions in hidden representations and then steer model activations accordingly (Li et al., 2023; von Rütte et al., 2024). Activation steering studies further suggest that middle-layer activations encode abstract attributes such as emotion, personality, and values, and that steering hidden states along these directions can reliably induce the corresponding behaviors (Tigges et al., 2024; Han et al., 2025; Jha et al., 2026). Decoding-based approaches have also been proposed; for example, Chuang et al. (2024) contrast layer-wise logits to amplify factual signals. However, these methods have not been systematically evaluated for reducing the knowledge-prediction gap.

In this paper, we bridge these two lines of research: we provide a geometric explanation of the knowledge-prediction gap and introduce KAPPA, an inference-time intervention that aligns model predictions with internal knowledge. We later show that KAPPA consistently reduces the gap and improves accuracy over existing intervention baselines.

# 3. Measuring Knowledge-Prediction Gap in MCQs

In this section, we demonstrate the **knowledge-prediction gap** in MCQ settings. We use linear probing to identify knowledge- and prediction-related signals in the model's hidden states and quantify the gap across benchmarks.

We define the **knowledge-prediction gap** as the discrepancy between the knowledge encoded in a model's internal representations and the predictions it ultimately generates. In this work, we focus on cases where this knowledge is **linearly accessible**—knowledge that can be extracted through simple linear operations on hidden states. While richer knowledge might be encoded in nonlinear forms, we focus on linear accessibility because it represents the most interpretable and

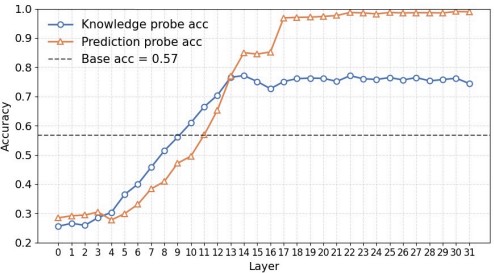

*Figure 2.* Layer-wise accuracy of the knowledge probe and the prediction probe on TruthfulQA for `Llama 3.1 8B Instruct`. While the knowledge probe accuracy increases steadily and saturates at higher layers, the prediction probe exhibits a sharper transition, lagging behind in earlier layers. The dashed line indicates the base model accuracy, highlighting a substantial knowledge-prediction gap in intermediate layers.

steerable form of knowledge. Furthermore, because linearly accessible information represents a relatively simple form of knowledge to exploit, its underutilization suggests that bridging internal knowledge and final predictions remains nontrivial. Importantly, linearly accessible knowledge is not necessarily leveraged and reflected in generation, leading to the knowledge-prediction gap—the problem we focus on.

## 3.1. Linear Probing Analysis

To detect knowledge- and prediction-related signals in hidden states, we train two linear probes on the model's residual stream: a *knowledge probe* that predicts the ground-truth answer, and a *prediction probe* that predicts the answer option chosen by the LLM. For an MCQ with $k$ options, each probe maps the hidden state to a $k$-dimensional vector, where each dimension corresponds to an MCQ option. When the knowledge probe correctly identifies the ground-truth answer, we treat the model as possessing the corresponding knowledge-signal in linearly accessible form.

Concretely, at each layer $l$, we first extract the residual stream activation $h^l(x) \in \mathbb{R}^d$ at the final token of the answer prefix and construct two datasets: $D_{\text{know}}^{(l)} = \{(h^l(x), y)\}$ pairing activations with ground-truth labels $y \in \{1, \ldots, k\}$, and $D_{\text{pred}}^{(l)} = \{(h^l(x), \tilde{y})\}$ pairing activations with the model's *greedy-decoded predictions* $\tilde{y} \in \{1, \ldots, k\}$. We then train a $k$-class linear classifier on each dataset to obtain the knowledge and prediction probes at every layer. See Appendix A.1 for probe selection.

**Setup.** We use `Llama 3.1 8B Instruct` and `Qwen2.5 7B Instruct` as representative open-weight models. Evaluation is conducted on a diverse suite of multiple-choice benchmarks including BBH (algorithmic and linguistic subtasks) (Suzgun et al., 2023), MMLU (STEM, Humanities, and Social Sciences) (Hendrycks et al., 2021), ARC-Challenge (Clark et al., 2018), TruthfulQA

*Table 1.* Model- and dataset-wise comparison of the knowledge-prediction gap for `Qwen2.5 7B Instruct` and `Llama 3.1 8B Instruct`. We report task accuracy of the base model (ACC), the absolute accuracy gain of the knowledge probe over the base model ($\Delta$ACC), and knowledge-prediction gap metrics (AGR and KLD). Higher $\Delta$ACC and KLD and lower AGR indicate larger knowledge-prediction gaps. The value $k$ denotes the number of answer options for each MCQ benchmark. Gray shading highlights the three largest knowledge-prediction gaps within each column.

| Category | Dataset | $k$ | Qwen2.5 7B | | | | Llama 3.1 8B | | | |
|---|---|---|---|---|---|---|---|---|---|---|
| | | | ACC | $\Delta$ACC | AGR | KLD | ACC | $\Delta$ACC | AGR | KLD |
| **Reasoning** | GSM8k | 4 | 47.8 | +2.5 | 60.3 | 0.86 | 32.6 | +4.1 | 53.7 | 0.35 |
| | BBH-Algorithmic | 4 | 51.0 | +4.4 | 69.6 | 0.70 | 45.1 | +5.5 | 62.1 | 0.23 |
| | BBH-NLP | 4 | 61.1 | +5.1 | 69.8 | 0.92 | 59.6 | +3.9 | 73.9 | 0.41 |
| **Knowledge** | MMLU Humanities | 4 | 59.9 | +2.6 | 78.5 | 0.78 | 58.6 | +3.4 | 77.9 | 0.47 |
| | MMLU Social Sciences | 4 | 78.8 | +0.2 | 95.9 | 0.72 | 74.0 | -0.6 | 90.5 | 0.47 |
| | MMLU STEM | 4 | 65.2 | +0.2 | 89.7 | 0.70 | 54.7 | +0.2 | 91.1 | 0.32 |
| | ARC-Challenge | 3 | 90.9 | +0.0 | 98.5 | 0.55 | 85.0 | +0.2 | 98.0 | 0.43 |
| | PubMedQA | 3 | 72.3 | -0.3 | 89.8 | 0.57 | 75.7 | +0.0 | 96.4 | 0.43 |
| **Truthfulness & Bias** | TruthfulQA | 4 | 58.8 | +21.3 | 61.8 | 1.01 | 56.7 | +19.6 | 62.1 | 0.63 |
| | BBQ-Age | 3 | 83.2 | +9.3 | 84.0 | 0.68 | 59.9 | +29.3 | 59.2 | 0.59 |
| | BBQ-Religion | 3 | 79.6 | +0.7 | 98.5 | 0.54 | 67.6 | +10.6 | 79.1 | 0.42 |

(Mahaut et al., 2024), GSM8k (Cobbe et al., 2021; Li et al., 2024), PubMedQA (Jin et al., 2019), and BBQ Age and Religion (Parrish et al., 2022). For BBH and TruthfulQA, we sample four choices per question due to varying numbers of answer choices per instance. Prompt formats are provided in Appendix A.2, and a complete list of subtasks and dataset statistics is provided in Appendix A.3.

**Quantitative Metrics.** Since accuracy (ACC) alone does not quantify magnitude of the knowledge-prediction gap in a way that allows comparison across benchmarks and models, we introduce two complementary metrics. Let $p_K(x), p_M(x) \in \mathbb{R}^k$ denote probability distributions over the $k$ answer options induced by the *knowledge probe* and the model for input $x$, respectively. The $i$-th entries $\big(p_K(x)\big)_i$ and $\big(p_M(x)\big)_i$ represent the probabilities assigned to the $i$-th option. Both distributions are obtained by applying a softmax over option-level logits.

**(1) Prediction Agreement** We measure decision-level alignment by checking whether the two distributions select the same option:

$$\text{AGR}(x) = \mathbb{I}\left[\arg\max_{i \in [k]}\big(p_K(x)\big)_i = \arg\max_{i \in [k]}\big(p_M(x)\big)_i\right].$$
(1)

Higher AGR indicates that the model's predictions more often match its encoded knowledge.

**(2) Distributional Divergence** Decision agreement does not capture differences in confidence across options. We therefore measure distribution-level alignment using Kullback-Leibler divergence:

$$\text{KLD}(x) = \text{KL}\big(p_M(x) \,\|\, p_K(x)\big).$$
(2)

Lower KLD indicates closer alignment between the model's output distribution and the knowledge probe's distribution.

Note that KLD is comparable only across benchmarks with the same number of options $k$.

**Results.** Table 1 shows the performance of the selected knowledge probes on each test set. The knowledge probes consistently outperform or match the LLMs' generation accuracy, with positive accuracy gains ($\Delta$ACC) observed across both model families and benchmarks. The layer-wise analysis (Figure 2; more results provided in Appendix A.4) reveals that this advantage emerges after the middle layers, where knowledge probe accuracy increases sharply. These results confirm that models encode linearly accessible knowledge that they fail to use in their final predictions. Prediction probes achieve higher accuracy than knowledge probes, often exceeding 90%, indicating that the model's eventual output is also strongly represented in hidden states. Together, these findings establish that hidden states robustly encode both the correct answer and the model's generated answer—yet these two signals diverge.

The magnitude of the gap varies substantially across domains: the gap is substantially larger for **truthfulness & bias** and **reasoning** benchmarks, whereas it remains comparatively small for **knowledge** benchmarks. For truthfulness & bias tasks, we hypothesize that models override their encoded knowledge with a preference for hasty answering, providing definite answers even when abstention is appropriate (Yang et al., 2024; Wen et al., 2024). For example, on BBQ, where many questions (∼50%) require selecting *"undetermined"*, `Llama 3.1 8B Instruct` chooses this option in only 13–27% of cases (accuracy: 59.9–67.6%), while knowledge probes achieve 38–51% (accuracy: 78.1–89.1%). For reasoning tasks, we hypothesize that models trained to generate intermediate traces may not fully leverage their internal representations when producing a final answer directly (Ma et al., 2025; Sun et al., 2025b). In both

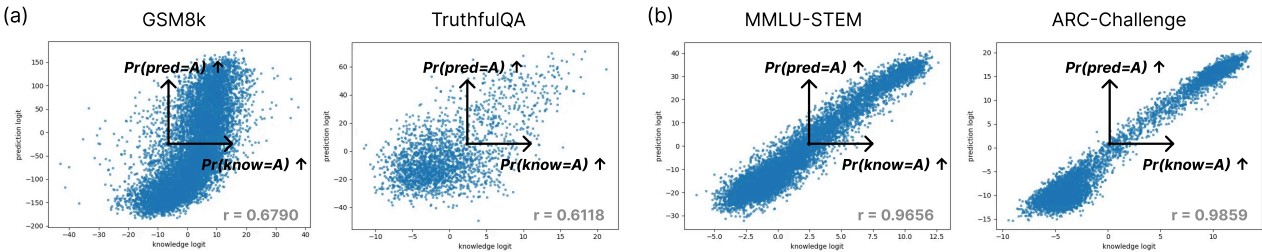

*Figure 3.* Scatter plots of hidden-state activations from `Qwen2.5 7B Instruct`, where each point corresponds to a pair of logits: the *knowledge logit* and the *prediction logit* for the first answer option. Results are shown for GSM8k (27th layer), TruthfulQA (19th layer), MMLU-STEM (23rd layer), and ARC-Challenge (24th layer). $r$ denotes the Pearson correlation between the two logits.

cases, hidden states encode correct answers that are not reflected in predictions. These results suggest that reducing the gap holds promise for mitigating hallucinations and improving accuracy on reasoning tasks.

# 4. Geometric Interpretation of the Gap

To better understand the representation geometry underlying the gap, we analyze the model's residual stream and the probes. First, we visualize knowledge and prediction probe logits for individual hidden-state representations (§4.1). Second, we analyze the geometric misalignment between the two probe-defined subspaces (§4.2). Finally, we test whether this misalignment is positively associated with the measured knowledge-prediction gap (§4.3).

**Setup.** For probes trained on a dataset with $k$-option MCQs, we treat each probe's weight matrix $W \in \mathbb{R}^{d \times k}$ as defining a subspace of the residual stream spanned by its columns $\{w_i\}_{i=1}^k$, yielding the *knowledge subspace* $\mathrm{span}(W_{\mathrm{know}})$ and *prediction subspace* $\mathrm{span}(W_{\mathrm{pred}})$. The probe logits $W^\top h$ (we omit the bias term for clarity) then represent the projection of the hidden state $h$ onto this subspace— its coordinate within that probe-defined subspace, where a larger value indicates higher probability for the corresponding MCQ option. We thus interpret the knowledge subspace as reflecting the model's internal confidence in which option is correct, and the prediction subspace its confidence in the option it predicts.

## 4.1. Visualizing Hidden State Activations

We first examine whether the gap manifests as misalignment in individual hidden states. For each hidden representation, we calculate the knowledge and prediction probe logits and plot them jointly for the first answer option. Ideally, a hidden state's coordinates in the two subspaces should align, but as we show below, they diverge when the gap is large.

**Results.** The benchmarks with a large knowledge-prediction gap (Section 3), such as GSM8k and TruthfulQA, exhibit substantial misalignment between the two logits

(Figure 3a), whereas benchmarks with a smaller gap, such as MMLU-STEM and ARC-Challenge, show much stronger alignment (Figure 3b). These results demonstrate that the knowledge-prediction gap is reflected in the residual stream geometry, motivating our approach of aligning the two coordinates to mitigate the gap, later described in Section 5.

## 4.2. Quantifying Subspace Misalignment

We now quantify subspace misalignment using two complementary tools from the representation alignment literature. Mean principal angle provides a dataset-independent measure of geometric separation between the two subspaces, while orthogonal linear CKA measures the representation-level similarity of hidden states after projection onto these subspaces. We compare both against matched random subspace baselines (mean principal angle $\approx 88.7°$, CKA $\lesssim 0.2$). We use three benchmarks with the largest knowledge-prediction gaps, measured by $1 - \mathrm{AGR}$: TruthfulQA, GSM8k, and BBH-Algorithmic (§3). More details are presented in Appendix B.

**Results.** Across layers, the mean principal angle rises from $60°$–$80°$ in the middle layers to nearly $90°$ in the later layers, approaching the random baseline (Figure 8). CKA, by contrast, remains in an intermediate range (0.4–0.8) throughout—well above random, but far short of full alignment (Figure 9). Together, these reveal that the knowledge and prediction signals coexist in the same residual stream while being routed along geometrically distinct directions.

## 4.3. Connecting the Knowledge-Prediction Gap to Subspace Misalignment

We next ask whether the observed geometric misalignment is merely a representational artifact, or whether it reflects the measured knowledge-prediction gap. To this end, across eight benchmarks, we pair the gap, defined as $1 - \mathrm{AGR}$ (§3), with the mean principal angle measured at the same layer, and compute Spearman's rank correlation (Appendix B).

**Results.** The result reveals a strong benchmark-level correspondence between subspace geometry and the gap. The

mean principal angle closely tracks the measured gap for `Llama 3.1 8B` ($\rho = 0.976$, $p = 0.001$), and `Qwen2.5 7B` shows a similar positive trend ($\rho = 0.619$, $p = 0.106$; Figure 10). This suggests that subspace misalignment is a systematic correlate of the gap: benchmarks with larger knowledge-prediction gaps tend to exhibit larger misalignment between the knowledge and prediction subspaces.

## 5. Bridging the Knowledge-Prediction Gap

To mitigate the knowledge-prediction gap in MCQs, we introduce KAPPA (**K**nowledge-**A**ligned **P**rediction through **P**rojection-based **A**djustment), an inference-time intervention that applies affine transformations to the residual stream. Motivated by the observed misalignment between the knowledge and prediction subspaces, KAPPA aligns the model's prediction with the knowledge encoded in the hidden states through a simple geometric correction.

### 5.1. Knowledge-Aligned Prediction Adjustment

Given an input prompt $x$ containing a multiple-choice question, KAPPA extracts the residual stream activation $h = h^l(x)$ at a layer $l$ for the last token of the prompt. It then computes the geometric coordinates of $h$ within the knowledge and prediction subspaces, as identified by the two linear probes. KAPPA minimally adjusts the hidden state within the prediction subspace so that its coordinates align with those in the knowledge subspace. This intervention is applied at each decoding step from the last token of the prompt to all subsequently generated tokens.

We formalize this intervention as a constrained optimization problem that adjusts hidden states to align prediction coordinates with corresponding knowledge coordinates, while remaining close to the original representations. For a hidden state $h$, we derive a transformation $\mathcal{T} : \mathbb{R}^d \to \mathbb{R}^d$ that maps $h$ to a modified state $h' = \mathcal{T}(h)$ under the following constraints: (1) *Alignment constraint*: the modified prediction coordinates must align with the knowledge coordinates; (2) *Minimal Perturbation*: the perturbation $\|h' - h\|$ is minimized so as to preserve other information.

Given the probe weights $W_{\text{pred}}, W_{\text{know}} \in \mathbb{R}^{d \times k}$ and biases $b_{\text{pred}}, b_{\text{know}} \in \mathbb{R}^{k \times 1}$, we derive the hidden-state update by solving the following constrained optimization problem:

$$\min_{h'} \ \|h' - h\|_2^2 \qquad \text{s.t.} \qquad \tilde{W}_{\text{pred}}^\top \tilde{h}' = \tilde{W}_{\text{know}}^\top \tilde{h}, \quad (3)$$

where we use augmented notation;

$$\tilde{h}' = \begin{bmatrix} h' \\ 1 \end{bmatrix}, \quad \tilde{h} = \begin{bmatrix} h \\ 1 \end{bmatrix},$$

$$\tilde{W}_{\text{know}} = \begin{bmatrix} W_{\text{know}} \\ b_{\text{know}}^\top \end{bmatrix}, \quad \tilde{W}_{\text{pred}} = \begin{bmatrix} W_{\text{pred}} \\ b_{\text{pred}}^\top \end{bmatrix} \in \mathbb{R}^{(d+1) \times k}.$$

Solving Eq. 3 yields a closed-form solution corresponding to the minimal $\ell_2$ modification that aligns the prediction coordinates with the knowledge coordinates (full derivation provided in Appendix C.1):

$$h' = h + W_{\text{pred}}(W_{\text{pred}}^\top W_{\text{pred}})^{-1} \left( \widetilde{W}_{\text{know}}^\top \tilde{h} - \widetilde{W}_{\text{pred}}^\top \tilde{h} \right).$$

This update minimally modifies the hidden state within the prediction subspace, leaving orthogonal components unchanged, while aligning its prediction coordinates with the corresponding knowledge coordinates. We apply KAPPA primarily at intermediate layers, where hidden states encode rich internal knowledge while still exerting substantial influence on downstream predictions. Unlike typical steering methods that apply a fixed direction across inputs, KAPPA dynamically computes an input-specific minimal adjustment based on the knowledge and prediction coordinates.

**Extended Alignment.** While the original alignment constraint in Eq. 3 enforces strict equality between the knowledge and prediction coordinates, we extend our method to capture more diverse knowledge-prediction relationships. We introduce hyperparameters $\alpha, \beta \in \mathbb{R}$:

$$\tilde{W}_{\text{pred}}^\top \tilde{h}' = \alpha \cdot \tilde{W}_{\text{know}}^\top \tilde{h} + \beta \cdot \text{sign}(\tilde{W}_{\text{know}}^\top \tilde{h}). \quad (4)$$

The original constraint is recovered when $\alpha = 1$ and $\beta = 0$, while larger values of $\alpha$ and $\beta$ amplify the influence of internal knowledge on model predictions. These hyperparameters provide two complementary controls: $\alpha$ sharpens confidence *across* MCQ options by amplifying relative differences in the knowledge logits, while $\beta$ sharpens confidence *within* each option by pushing prediction probe logits toward positive or negative extremes.

### 5.2. Experiments

**Setup.** Using the identified knowledge and prediction subspaces for each dataset, we evaluate KAPPA at inference time on six diverse benchmarks showing a large knowledge-prediction gap in Section 3. Furthermore, to examine whether KAPPA generalizes across model architectures and sizes, we additionally evaluate KAPPA on `Mistral 7B Instruct v0.3`, `Qwen3 4B`, and `Qwen3 14B` using the TruthfulQA dataset. For KAPPA intervention, we select the top-$n$ consecutive layers with the highest knowledge probe accuracies from those where the prediction probe accuracy exceeds a threshold. Details of the evaluation protocol are provided in Appendix A.1.

To compare KAPPA's effectiveness with prior inference-time methods in reducing the knowledge-prediction gap, we consider the following baselines: (i) **Base**: the original LLM; (ii) **CAA**: activation steering using difference-in-means vectors (Rimsky et al., 2024); (iii) **DoLA**: a decoding method

*Table 2.* Results across benchmarks for `Qwen2.5 7B Instruct` and `Llama 3.1 8B Instruct`. Higher ACC indicates better performance, and higher AGR and lower KLD indicate closer knowledge-prediction alignment. Benchmark names annotated with (·) denote the number of answer options in the corresponding MCQ setting. The KP rows report results computed directly from the knowledge probe's predictions. For KAPPA(·), the value in parentheses denotes the number of layers intervened at inference time. Boldface indicates the best result among {Base, CAA, DoLA, KAPPA(·)}, excluding the KP row.

| Model | Method | GSM8k (4) | | | BBH-Algo (4) | | | BBH-NLP (4) | | | TruthfulQA (4) | | | BBQ-Age (3) | | | BBQ-Religion (3) | | |
|---|---|---|---|---|---|---|---|---|---|---|---|---|---|---|---|---|---|---|---|
| | | ACC | AGR | KLD | ACC | AGR | KLD | ACC | AGR | KLD | ACC | AGR | KLD | ACC | AGR | KLD | ACC | AGR | KLD |
| Qwen2.5 7B | Base | 47.8 | 60.3 | 0.86 | 51.0 | 69.6 | 0.70 | 61.1 | 69.8 | 0.92 | 58.8 | 61.8 | 1.01 | 83.2 | 84.0 | 0.68 | 79.6 | 98.5 | **0.54** |
| | CAA | 47.7 | 60.3 | 0.85 | 51.2 | 69.8 | 0.71 | 61.2 | 69.9 | 0.92 | 61.1 | 63.5 | 0.98 | 84.3 | 85.0 | 0.67 | 79.5 | 98.5 | 0.54 |
| | DoLA | 48.1 | 60.0 | 0.86 | 50.4 | 67.5 | 0.75 | 61.0 | 69.3 | 0.92 | 58.5 | 61.3 | 1.02 | 82.9 | 83.9 | 0.68 | 79.5 | 98.2 | 0.54 |
| | KAPPA (1) | **49.6** | **68.8** | 0.78 | 51.5 | 72.5 | 0.69 | 63.0 | 74.0 | 0.89 | 60.6 | 64.0 | 0.99 | 84.1 | 85.2 | 0.67 | 80.1 | **99.4** | 0.55 |
| | KAPPA (3) | 49.1 | 65.4 | 0.77 | 52.2 | 75.3 | **0.65** | 62.8 | 73.2 | 0.89 | 61.9 | 65.1 | 0.97 | 83.9 | 84.9 | 0.67 | 80.2 | 99.0 | 0.55 |
| | KAPPA (6) | 49.2 | 66.3 | **0.76** | **53.6** | **78.9** | 0.66 | **63.6** | **74.9** | **0.87** | **64.1** | **67.3** | **0.95** | **85.5** | **87.0** | **0.64** | **80.5** | 98.8 | 0.55 |
| | KP | 50.3 | 100.0 | 0.00 | 55.4 | 100.0 | 0.00 | 66.2 | 100.0 | 0.00 | 80.1 | 100.0 | 0.00 | 92.5 | 100.0 | 0.00 | 80.3 | 100.0 | 0.00 |
| Llama 3.1 8B | Base | 32.6 | 53.7 | 0.35 | 45.1 | 62.1 | **0.23** | 59.6 | 73.9 | 0.41 | 56.7 | 62.1 | 0.63 | 59.9 | 59.2 | 0.59 | 67.6 | 79.1 | 0.42 |
| | CAA | 32.9 | 53.8 | 0.38 | 45.1 | 62.1 | 0.26 | 60.2 | 73.3 | 0.41 | 62.3 | 67.2 | 0.60 | 65.8 | 64.8 | 0.53 | 68.8 | 80.0 | 0.42 |
| | DoLA | 33.2 | 49.7 | 0.58 | 42.4 | 47.4 | 0.29 | 56.6 | 69.6 | 0.51 | 55.6 | 61.6 | 0.76 | 59.6 | 60.1 | 0.64 | 64.9 | 76.6 | 0.42 |
| | KAPPA (1) | 34.9 | 73.8 | 0.37 | 49.3 | **83.1** | 0.31 | 62.4 | **86.3** | 0.47 | 67.8 | **78.8** | 0.60 | 73.8 | 75.0 | 0.45 | 75.7 | **93.6** | 0.46 |
| | KAPPA (3) | 34.6 | 66.2 | 0.27 | 49.5 | 82.9 | 0.26 | **63.0** | 83.7 | **0.39** | 65.6 | 72.9 | 0.48 | 72.3 | 74.4 | 0.38 | **76.5** | 92.9 | **0.41** |
| | KAPPA (6) | **36.6** | 75.9 | **0.27** | **50.1** | 82.5 | 0.30 | 60.2 | 75.3 | 0.46 | **73.5** | 77.6 | **0.46** | **76.8** | **81.1** | **0.31** | 68.8 | 81.6 | 0.57 |
| | KP | 36.6 | 100.0 | 0.00 | 50.7 | 100.0 | 0.00 | 63.5 | 100.0 | 0.00 | 76.3 | 100.0 | 0.00 | 89.1 | 100.0 | 0.00 | 78.1 | 100.0 | 0.00 |

that contrasts layer-wise logits (Chuang et al., 2024). More details are provided in Appendix D.

**Results Across Datasets.** Table 2 demonstrates the effectiveness of KAPPA across six diverse benchmarks spanning reasoning, truthfulness, and bias domains, where substantial knowledge-prediction gaps are observed. As shown in the AGR column, KAPPA consistently improves over base models and prior inference-time methods by significantly increasing agreement between the knowledge probe and the model's final predictions (up to 22.2% points over base). Crucially, these improved agreements translate into substantial gains in accuracy, often approaching the accuracy of the knowledge probe (KP), which is achieved when the model fully utilizes its linearly accessible knowledge. The gains are especially pronounced in truthfulness and bias benchmarks: for `Llama 3.1 8B`, KAPPA achieves improvements of 8.9–16.9 points on TruthfulQA and BBQ, while delivering consistent gains of 1.5–5.0 points in reasoning benchmarks. Conversely, on knowledge benchmarks where the knowledge-prediction gap is negligible, KAPPA yields only modest changes (see Appendix E.1). This confirms that our method mitigates the misalignment between internal knowledge and output prediction.

**Results Across Models.** Table 3 indicates that the knowledge-prediction gap is a pervasive phenomenon across a broad range of model families and sizes. Although larger-scale models tend to exhibit a relatively narrower gap compared to smaller counterparts (AGR: 76% for `Qwen3 14B` vs. 60% for `Qwen3 4B`), the misalignment remains substantial. KAPPA consistently mitigates this gap and improves accuracy across all evaluated models, supporting our hypothesis that many model errors arise from a geometric misalignment between internal knowledge and output predictions. Additional results on `Qwen3 32B` further

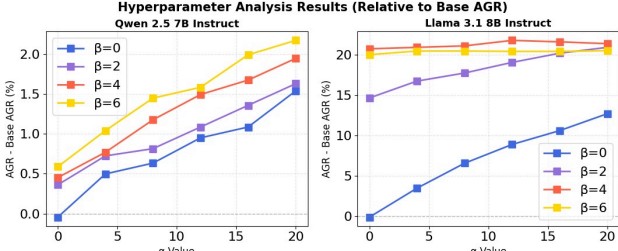

*Figure 4.* Effect of parameters $\alpha$ and $\beta$ on agreement rate improvement over the base model. (**Left**) `Qwen2.5 7B Instruct` results across six $\alpha$ values and four $\beta$ values. (**Right**) `Llama 3.1 8B Instruct` results under the same settings.

demonstrate that KAPPA remains effective in mitigating the knowledge-prediction gap even at larger scales. These findings suggest the gap is a fundamental phenomenon across diverse model scales and training strategies, spanning both distilled and non-distilled models, with further interpretations detailed in Appendix E.2.

**Hyperparameter Sweep.** To study the effects of the parameters $\alpha$ and $\beta$ in Eq. 4, we perform a sweep on TruthfulQA, and find that increasing either parameter improves agreement (Figure 4). Larger values amplify the prediction coordinate, thereby steering model outputs toward the knowledge-aligned answer. These results suggest that both parameters causally control the strength of alignment.

**Sensitivity Analysis.** To examine whether KAPPA is robust to the construction of the probe training dataset, we conduct two analyses: (i) sensitivity to MCQ distractor sampling randomness and (ii) sensitivity to dataset size. Detailed experimental setups and results are provided in Appendix E.3.

For distractor sampling, we reuse the probes trained in the main experiments and evaluate KAPPA on test sets con-

*Table 3.* Average accuracy (ACC), agreement (AGR), and KL divergence (KLD) across multiple models on the TruthfulQA benchmark. For KAPPA(·), the value in parentheses denotes the number of layers intervened at inference time. Boldface indicates the best result among {Base, CAA, DoLA, KAPPA(·)}, excluding the knowledge probe (KP).

| Method | Mistral v0.3 7B | | | Llama-3.1 8B | | | Qwen2.5 7B | | | Qwen3 4B | | | Qwen3 14B | | |
|---|---|---|---|---|---|---|---|---|---|---|---|---|---|---|---|
| | ACC | AGR | KLD | ACC | AGR | KLD | ACC | AGR | KLD | ACC | AGR | KLD | ACC | AGR | KLD |
| Base | 40.7 | 46.6 | 0.94 | 56.7 | 62.1 | 0.63 | 58.8 | 61.8 | 1.01 | 56.5 | 60.0 | 0.82 | 71.6 | 76.0 | 0.76 |
| CAA | 52.8 | 55.9 | 0.83 | 62.3 | 67.2 | 0.60 | 61.1 | 63.5 | 0.98 | 60.1 | 63.2 | 0.83 | 73.6 | 77.9 | 0.76 |
| DoLA | 40.5 | 46.7 | 0.94 | 55.6 | 61.6 | 0.76 | 58.5 | 61.3 | 1.02 | 56.5 | 59.4 | 0.83 | 71.3 | 75.9 | 0.77 |
| KAPPA (1) | 51.0 | 59.7 | 0.82 | 67.8 | **78.8** | 0.60 | 60.6 | 64.0 | 0.99 | 58.4 | 62.3 | 0.79 | 73.3 | 78.1 | 0.76 |
| KAPPA (3) | 56.3 | **64.1** | 0.68 | 65.6 | 72.9 | 0.48 | 61.9 | 65.1 | 0.97 | 58.9 | 62.6 | 0.75 | 74.3 | 79.2 | 0.74 |
| KAPPA (6) | **58.3** | 62.3 | **0.65** | **73.5** | 77.6 | **0.46** | **64.1** | **67.3** | **0.95** | **61.4** | **66.1** | **0.70** | **77.7** | **83.7** | **0.72** |
| KP | 69.5 | 100.0 | 0.00 | 76.3 | 100.0 | 0.00 | 80.1 | 100.0 | 0.00 | 78.7 | 100.0 | 0.00 | 85.8 | 100.0 | 0.00 |

structed with different randomly sampled distractors across 10 random seeds. The standard deviation of performance remains small across settings (1.281–2.529), indicating that KAPPA's gains are not overly sensitive to distractor sampling (Table 14). One-sided $t$-tests further show that KAPPA significantly outperforms the base model in 7 out of 8 configurations, with near-zero $p$-values ($p < 0.0001$), confirming its robustness to distractor sampling variation (Table 15).

To evaluate sensitivity to probe training data size, we subsample the training set from 10% to 80% across three random seeds, retrain the probes, and apply KAPPA. Although performance generally improves with more training data, KAPPA outperforms the base model even with only 10% of the data in all evaluated settings, suggesting its effectiveness in low-data regimes (Figure 11). Detailed experimental setup and results are provided in Appendix E.3.

## 6. Analysis

Having shown that KAPPA reduces the knowledge-prediction gap and improves MCQ performance, we now ask how KAPPA exerts this effect. First, we test whether KAPPA operates through direct logit shifting and find that it does not (§6.1). We then examine whether its effect transfers across datasets (§6.2) and beyond the MCQ format (§6.3).

### 6.1. Comparing Prediction Subspace with Logit Space

To examine how KAPPA affects the model's prediction process, we test a simple mechanistic hypothesis: KAPPA may act by directly shifting the logits of the answer-symbol tokens favored by the knowledge probe. Concretely, we compare the prediction subspace with the logit subspace spanned by the rows of the unembedding matrix corresponding to the answer-symbol tokens, which directly determine their output logits. The mean principal angle between these two subspaces measures the extent to which KAPPA's intervention aligns with direct logit shifting.

We find that, although the two subspaces become more aligned with depth, they remain clearly separated at the

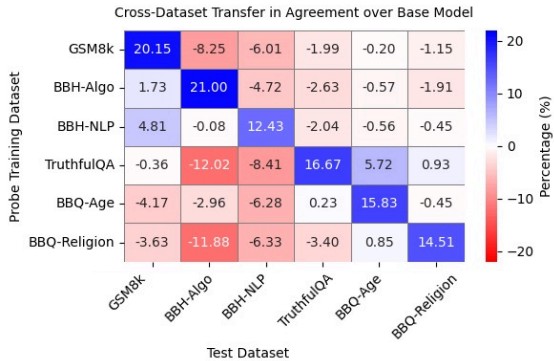

*Figure 5.* Cross-dataset transfer results measured by agreement improvement over the base model on `Llama 3.1 8B Instruct`. Positive values indicate improvement over base and negative values indicate degradation. Rows indicate the dataset used to train the probe, and columns indicate the dataset used for evaluation.

KAPPA intervention layers, with mean principal angles around $65°$–$70°$. This indicates that KAPPA does not primarily operate by directly modifying the logits of the answer-symbol tokens. Instead, KAPPA appears to influence the model's decision through more abstract intermediate representations, at layers where the representation is not yet strongly aligned with the final logit directions. Detailed setup and results are provided in Appendix B.6.

### 6.2. Transfer Across Datasets

We next examine whether this representation captures shared structure that transfers across datasets. To test this, we apply probes trained on one dataset to another and report improvements in agreement over the base model.

**Results.** As shown in Figure 5, many cross-dataset transfers show limited transferability, indicating that different datasets and tasks have geometrically distinct knowledge and prediction subspaces. Nevertheless, we observe successful transfer in some settings. These cases involve transfers within the same category, where the source and target datasets require broadly similar underlying skills. For example, applying

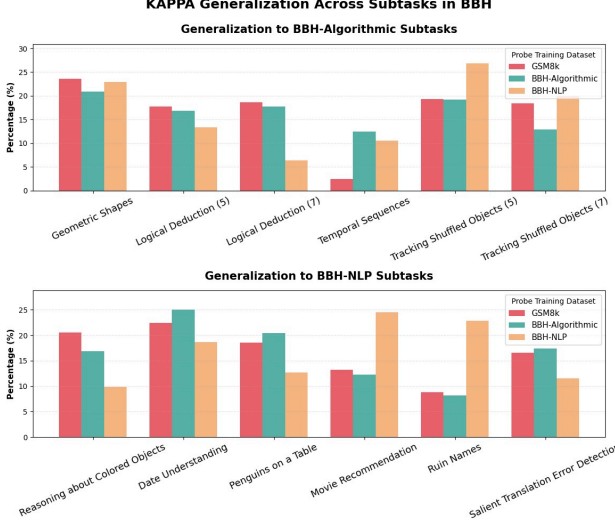

*Figure 6.* We present a per-subtask analysis of KAPPA generalization on BBH benchmarks. For each probe training dataset (GSM8k, BBH-Algorithmic, BBH-NLP), we identify questions that were originally answered incorrectly by the base model but become correct after applying KAPPA, and report the distribution of these improved questions across BBH subtasks.

subspaces extracted from BBH-NLP to GSM8k yields a 4.81% point gain, and transferring subspaces from TruthfulQA to BBQ-Age results in a 5.72% point gain. Notably, this transfer remains effective even though the two datasets use different answer symbols (TruthfulQA: 1/2/3/4; BBQ-Age: A/B/C) and different numbers of options ($k = 4 \rightarrow 3$). These results suggest that KAPPA does not operate by simply shifting answer-symbol logits, but instead operates on a more abstract internal representation.

**Subtask-level Analysis.** Focusing on transfer patterns into BBH subtasks, we find that subspace transfer is more effective between tasks that share similar underlying reasoning skills. Figure 6 shows the distribution of BBH subtasks for which test instances change from incorrect to correct after transferring subspaces extracted from different reasoning benchmarks. Here, BBH-Algorithmic primarily consists of subtasks requiring logical reasoning, whereas BBH-NLP emphasizes linguistic and commonsense understanding. When subspaces extracted from GSM8k or BBH-Algorithmic are applied to BBH-NLP, performance gains are concentrated on subtasks involving logical reasoning (e.g., *Reasoning about Colored Objects*, *Date Understanding*), rather than on linguistic or commonsense-oriented tasks (e.g., *Movie Recommendation*, *Ruin Names*). Extended results are presented in Appendix E.4.

### 6.3. Transfer to Free-Form Generation

We now evaluate whether KAPPA generalizes beyond MCQs to free-form generation. We prompt the model with

*Table 4.* Generation accuracy of `Qwen2.5 7B Instruct` after applying KAPPA to free-form settings.

| Method | TruthfulQA | BBQ-Age | GSM8k |
|---|---|---|---|
| Base | 41.7 | 89.7 | **91.6** |
| KAPPA (1) | **44.2** | **89.9** | 90.7 |

questions only—without MCQ options—and apply KAPPA at every token generation step, using subspaces extracted from the same dataset under the MCQ setting. For evaluation, we adopt an LLM-as-judge following Chandak et al. (2025), using `GPT-4o-mini` to compute accuracy.

**Results.** As shown in Table 4, applying KAPPA improves the model accuracy on TruthfulQA from 41.7% to 44.2%. This provides initial evidence that KAPPA, despite being trained only on MCQ data, can transfer to free-form generation. When the base model is already strong, KAPPA largely preserves performance: accuracy remains comparable on BBQ-Age (89.7% → 89.9%) and shows only a marginal drop on GSM8k (91.6% → 90.7%).

Across the three analyses (§6.1, §6.2, §6.3), the findings indicate that KAPPA modifies high-level, output-related features in the intermediate-layer prediction subspaces—features that later layers transform into concrete token predictions, whether MCQ answer symbols or free-form responses. This view is consistent with mechanistic interpretability work showing that hidden states encode high-level features (Park et al., 2024), which are progressively transformed into token predictions by later layers (nostalgebraist, 2020; Geva et al., 2023; Lad et al., 2026). It also aligns with activation steering studies showing that middle-layer directions can encode abstract attributes such as emotion, personality, and values, and causally induce corresponding behaviors (Tigges et al., 2024; Han et al., 2025; Jha et al., 2026).

## 7. Conclusion

This work investigates the knowledge-prediction gap in LLMs and shows that it is pervasive in diverse MCQ settings. Through a geometric analysis of the residual stream, we demonstrate that misalignment between distinct knowledge and prediction subspaces underlies many model errors. Building on this, we introduce KAPPA, which consistently reduces this gap and improves performance across truthfulness, bias, and reasoning benchmarks at inference time. These results suggest that many failures of LLMs stem not from missing knowledge, but from how the knowledge is utilized. Our findings highlight representation alignment as a promising approach to eliciting faithful model behavior. Several important extensions remain, including more reliably extracting knowledge from hidden states, aligning nonlinearly encoded knowledge, evaluating broader prompting settings, and extending to higher-dimensional interventions.

## Impact Statement

This paper presents work whose goal is to advance the understanding of internal representations in large language models and to improve the alignment between model knowledge and predictions. While improved reliability of model outputs may have downstream societal benefits, we do not anticipate significant negative societal impacts specific to this work.

## Acknowledgments

This work was supported by the National Research Foundation of Korea (NRF) grant RS-2024-00333484 and by the Institute of Information & Communications Technology Planning & Evaluation (IITP) grant (RS-2024-00338140, Development of Learning and Utilization Technology to Reflect Sustainability of Generative Language Models and Up-to-dateness over Time), both funded by the Korean government (MSIT). This research was supported by Basic Science Research Program through the National Research Foundation of Korea (NRF) funded by the Ministry of Education (RS-2025-25431623).

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

## A. Detailed Setup for the Knowledge-Prediction Gap

### A.1. Evaluation Pipeline

Our evaluation pipeline proceeds in four stages. **First**, we evaluate all prompt formats on the validation set and select the format yielding the most balanced predictions across answer options. This enables us to isolate the knowledge-prediction gap from the positional bias in multiple-choice questions. **Second**, we extract activations from all model layers and train both the knowledge probe and the prediction probe. For probe training, we use a batch size of 128, learning rate of 0.0002, and select the checkpoint with the best validation accuracy across 1000 epochs. **Third**, to select effective layers for our evaluation, we select layers whose prediction probe accuracy exceeds 85% on the validation set, and then identify the top $n$ consecutive layers based on knowledge probe performance on the same validation set. **Finally**, we apply 1-layer, 3-layer, and 6-layer KAPPA to the test datasets. Without model- or dataset-specific hyperparameter tuning, we use consistent settings across all configurations: $\alpha = 30, \beta = 0$ for single-layer interventions and $\alpha = 10, \beta = 0$ for multi-layer interventions.

### A.2. Prompt Formats and Example

To mitigate the effects of option bias—where the model favors a particular choice (Zheng et al., 2024; Pezeshkpour & Hruschka, 2024)—we evaluate the base model across multiple prompt formats (32 combinations), derived from four alternative instructions, two answer formats, and four option symbols, and select the format that yields the most balanced predictions on the validation set. We present each component in Table 5, Table 6, and Table 7.

*Table 5.* Task instructions used in our experiments.

| Instruction | Text |
|---|---|
| 1 | Provide your answer with {option_text}, following the output format below. **Output Format:** {format_text} |
| 2 | Given the following question and candidate answers, choose the correct option. **Output Format:** {format_text} |
| 3 | Choose the correct option between {option_text}. Use the format shown below when you answer. **Output Format:** {format_text} |
| 4 | Choose the best answer. Follow the output format below. **Output Format:** {format_text} |

*Table 6.* Answer format types used in experiments.

| Answer Format Type | Format Text |
|---|---|
| 1 | The correct answer is ( |
| 2 | Answer: ( |

*Table 7.* Option symbol types used in experiments.

| Option Symbol Type | Examples |
|---|---|
| Common Alphabet Upper | (A), (B), (C), (D) |
| Roman Numerals | (i), (ii), (iii), (iv) |
| Common Numbers | (1), (2), (3), (4) |
| Random Alphabet Upper | (Q), (K), (E), (U) |

We present one example of the prompt format used in our experiments, shown below.

---

**Multiple-Choice Prompt**

**Instruction.** Choose the correct option between (1), (2), (3), or (4). Use the format shown below when you answer.

**Output Format.**

```
The correct answer is (1) or
The correct answer is (2) or
The correct answer is (3) or
The correct answer is (4)
```

**Question.**
In Python 3, which of the following functions converts a string to an integer?

**Choices.**

(1) `float(x)`

(2) `int(x[, base])`

(3) `long(x[, base])`

(4) `str(x)`

---

## A.3. Data Statistics and Construction

### A.3.1. DATASET SIZES AND SUBTASKS

**Dataset Sizes.** Table 8 summarizes the number of samples in each split and the number of answer choices for each dataset. Each question is evaluated under multiple option orders. This procedure neutralizes positional biases while preserving exact class balance across the entire dataset.

*Table 8.* Dataset statistics for training, validation, and test splits. $k$ denotes the number of answer choices.

| Category | Dataset | k | Train | Validation | Test |
|---|---|---|---|---|---|
| Reasoning | GSM8k | 4 | 8960 | 2992 | 10552 |
| | BBH-Algorithmic | 4 | 4800 | 2400 | 4800 |
| | BBH-NLP | 4 | 4464 | 2232 | 4472 |
| Knowledge | MMLU Humanities | 4 | 15000 | 7536 | 15104 |
| | MMLU Social Sciences | 4 | 9800 | 4936 | 9880 |
| | MMLU STEM | 4 | 10040 | 5064 | 10120 |
| | ARC-Challenge | 3 | 6714 | 1794 | 7032 |
| | PubMedQA | 3 | 2400 | 600 | 3000 |
| Truthfulness & Bias | TruthfulQA | 4 | 2208 | 1104 | 2208 |
| | BBQ-Age | 3 | 8832 | 4416 | 8832 |
| | BBQ-Religion | 3 | 2880 | 1440 | 2880 |

**Subtasks.** For BBH and MMLU, we construct evaluation sets based on the subtask compositions defined in the original benchmark papers. In the case of BBH, we exclude subtasks that contain only two or three options or cannot be naturally formatted as MCQs, retaining only those with four or more options that support unambiguous multiple-choice formatting. The complete list of included subtasks for BBH and MMLU is provided in Table 9 and Table 10, respectively.

*Table 9.* Subtasks used from the BBH-Algorithmic and BBH-NLP benchmarks.

| Dataset | BBH-Algorithmic | BBH-NLP |
|---|---|---|
| Subtasks | geometric shapes | date understanding |
| | logical deduction | penguins in a table |
| | temporal sequences | movie recommendation |
| | tracking shuffled objects | ruin names |
| | reasoning about colored objects | salient translation error detection |

*Table 10.* Subtasks used from the MMLU Humanities, Social Sciences, and STEM benchmarks.

| Dataset | Subtasks | |
|---|---|---|
| **MMLU Humanities** | Formal Logic
High School European History
High School US History
High School World History
International Law
Jurisprudence
Logical Fallacies | Moral Disputes
Moral Scenarios
Philosophy
Prehistory
Professional Law
World Religions |
| **MMLU Social Sciences** | High School Geography
High School Gov't and Politics
High School Macroeconomics
High School Microeconomics
High School Psychology
Human Sexuality | Professional Psychology
Public Relations
Security Studies
Sociology
US Foreign Policy |
| **MMLU STEM** | Abstract Algebra
Anatomy
Astronomy
College Biology
College Chemistry
College Computer Science
College Mathematics
College Physics
Computer Security | Electrical Engineering
Elementary Mathematics
High School Biology
High School Chemistry
High School Computer Science
High School Mathematics
High School Physics
High School Statistics
Machine Learning |

### A.3.2. DATA CONSTRUCTION DETAILS

To adapt BBH and TruthfulQA to our experimental framework, we standardize all items to four-choice multiple-choice questions. For items originally containing more than four options, we randomly sample three incorrect options and combine them with the correct answer. This procedure ensures a consistent MCQ format while preserving the original label distribution. For GSM8k, which is not originally formulated as a multiple-choice benchmark, we use the multiple-choice version provided by Li et al. (2024).

### A.4. Additional Linear Probing Results

Figure 7 presents probing results across benchmarks using `Llama 3.1 8B Instruct`, showing that the probe trained to predict correct answers achieves stable performance across layers, consistently exceeding the base model accuracy.

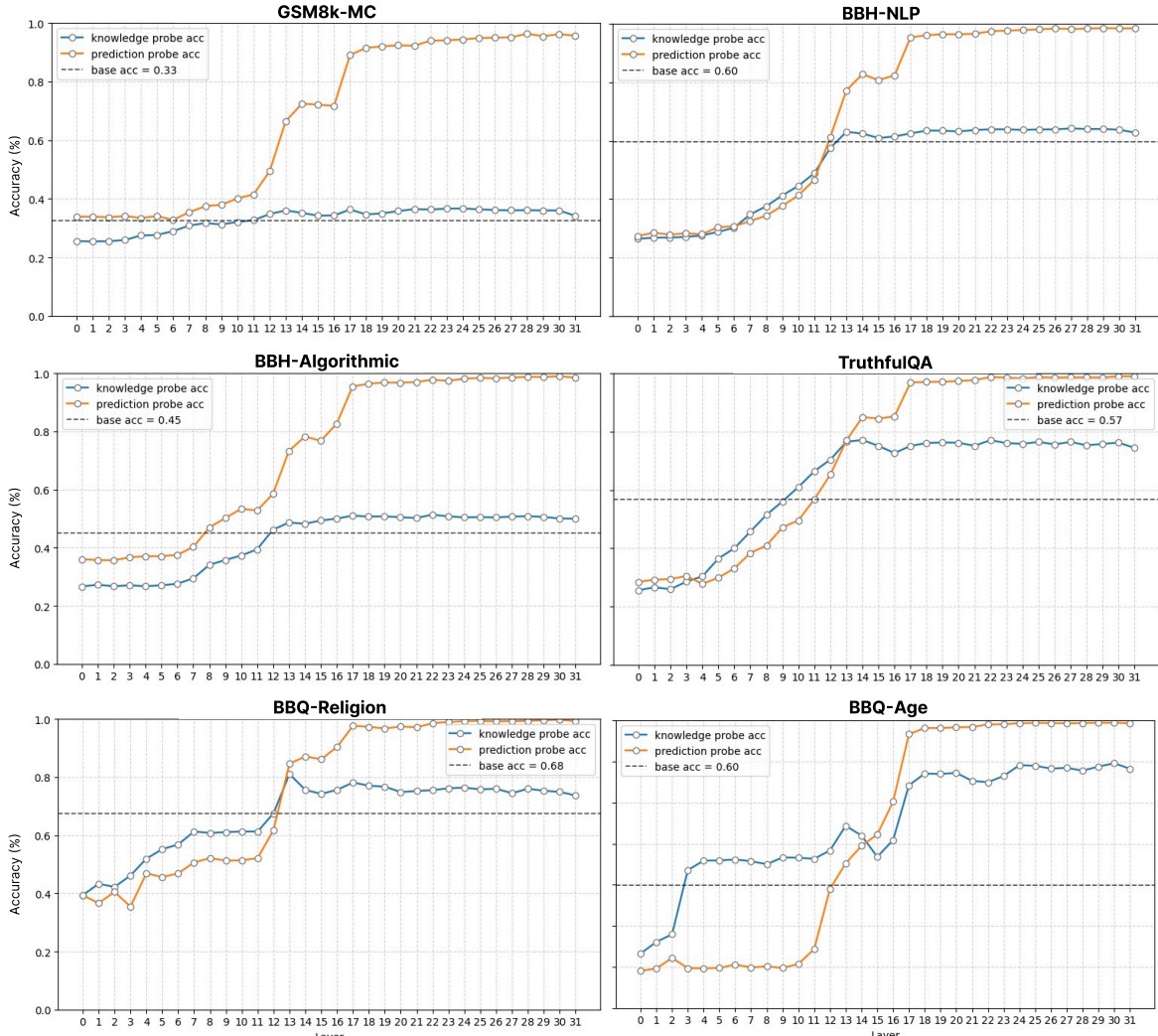

*Figure 7.* Layer-wise accuracy of knowledge probes (blue) and prediction probes (orange) across benchmarks, with baseline accuracy shown as dashed lines. Prediction probe accuracy reaches very high levels in later layers, and knowledge probe accuracy consistently exceeds baseline model accuracy, suggesting a significant knowledge-prediction gap.

# B. Detailed Setup for the Geometric Analysis

## B.1. Subspace Construction

For a $k$-option probe with weight matrix $W \in \mathbb{R}^{k \times d}$, adding the same vector to every row shifts all option logits by a common amount and therefore leaves the softmax/argmax decision unchanged. We therefore analyze the *decision-relevant* subspace, defined as the span of the mean-centered weights $\tilde{w}_i = w_i - \frac{1}{k} \sum_{j=1}^{k} w_j$. Concretely, for each probe, we construct an orthonormal basis $Q \in \mathbb{R}^{d \times r}$ from the left singular vectors of $\tilde{W}^\top$ whose singular values exceed the numerical tolerance $\epsilon = \text{eps} \cdot \max(k, d) \cdot \sigma_{\max}(\tilde{W})$, where $r \leq k - 1$ is the resulting rank. We apply the same construction to the knowledge probe ($W_{\text{know}}$), the prediction probe ($W_{\text{pred}}$), and all other subspaces considered.

## B.2. Subspace Alignment Metrics

**Mean Principal Angles.**    Mean principal angle measures span-level geometry. Given orthonormal bases $Q_A, Q_B$, the principal angles $0 \leq \theta_1 \leq \cdots \leq \theta_r \leq 90°$ are defined by

$$\cos\theta_j = \sigma_j(Q_A^\top Q_B),$$

the singular values of the basis overlap. Each angle records, one shared direction at a time, how closely the two subspaces line up: $\theta_j = 0°$ marks a direction common to both subspaces, while $\theta_j = 90°$ marks a direction in one that is orthogonal to the other. We report the mean over the $r$ angles as a single scalar, so that $0°$ means identical spans and $90°$ means fully orthogonal spans. This metric depends only on the subspaces themselves—it is invariant to the choice of basis and to the data distribution—and thus isolates pure geometry.

**Orthogonal Linear CKA.**    Orthogonal linear CKA measures example-level similarity of the representations that the two subspaces actually carry. We project the residual-stream activations onto each orthonormal basis,

$$X = HQ_A, \qquad Y = HQ_B,$$

where $H \in \mathbb{R}^{N \times d}$ stacks the per-example activations, and compute linear centered CKA

$$\text{CKA}(X, Y) = \frac{\|X_c^\top Y_c\|_F^2}{\|X_c^\top X_c\|_F \cdot \|Y_c^\top Y_c\|_F} \in [0, 1],$$

where $X_c, Y_c$ are mean-centered over examples. Intuitively, CKA asks whether the two subspaces order and separate the same examples in the same way; it reflects the joint distribution of the projected representations rather than the geometry of the spans alone. We call it *orthogonal* CKA because it is computed in the orthonormalized probe spans, making it invariant to how each probe is parameterized.

**Complementarity of the Two Metrics.**    The two metrics capture complementary aspects of subspace alignment: principal angles measure geometric separation between subspaces, whereas CKA detects shared structure in the projected representations across examples. Thus, even when two subspaces are nearly orthogonal, their projections can remain correlated due to anisotropy in the residual stream. Together, these metrics provide a more complete view of subspace misalignment.

## B.3. Random Baselines

We compare each against a baseline constructed from random subspaces of matched rank.

**Principal Angle Baseline.**    We draw $W \sim \mathcal{N}(0, 1)^{k \times d}$, mean-center its rows, and compute the mean principal angle between two such random subspaces. Averaged over 100 samples, the resulting baseline is $\approx 88.7°$. The baseline lies close to fully orthogonal because, in a $\sim$4000-dimensional residual stream, two low-rank ($r \leq k - 1$) subspaces drawn at random are nearly orthogonal by default.

**CKA Baseline.**    We draw a random $d \times r$ orthonormal basis $Q_{\text{rand}}$ (from the QR decomposition of a $d \times r$ Gaussian matrix), and compute linear centered CKA between $HQ$ (the probe-projected activations) and $HQ_{\text{rand}}$. Averaged over 32 samples with seed 1729, the resulting baseline is $\lesssim 0.2$.

We therefore interpret principal-angle values close to $\approx 88.7°$ as being near-orthogonal, and CKA values clearly above 0.2 as representations retaining similar structure beyond chance.

## B.4. Layer-wise Misalignment Results

We present the layer-wise behavior of the two geometric metrics on `Llama 3.1 8B Instruct` and `Qwen2.5 7B Instruct` across the three benchmarks with the largest knowledge-prediction gap identified in §3: TruthfulQA, GSM8k, and BBH-Algorithmic.

**Mean Principal Angle Across Layers.** As shown by the blue curves in Figure 8, the mean principal angle between the knowledge and prediction subspaces is moderate in early-to-middle layers (roughly $60°$–$80°$), suggesting partial overlap. At the late layers selected for KAPPA intervention (vertical lines), however, the angle rises sharply toward $90°$, nearly matching the random-subspace baseline ($\approx 88.7°$). Thus, precisely where the model forms its final answer, the directions encoding which option is correct become nearly as misaligned with the directions encoding which option will be selected as two random subspaces. This pattern is consistent across both model families and all three benchmarks.

**Orthogonal Linear CKA Across Layers.** Figure 9 shows that at the KAPPA intervention layers, the orthogonal CKA sits in an intermediate range (about $0.4$–$0.8$), clearly above the random control ($\lesssim 0.2$) yet well short of full alignment. Read together with the principal-angle result, this paints a coherent picture: because the knowledge- and prediction-projected features are derived from the same residual stream, they retain some example-level similarity (CKA stays nonzero), but the spans along which the two signals live are largely disjoint (angles near $90°$). The intermediate CKA does not contradict the near-orthogonal angle—the two metrics measure different things, and a highly anisotropic residual stream can keep two projections correlated even when their spans barely overlap.

**Takeaway.** The model encodes both the correct-answer signal and its eventual prediction in the hidden state, but routes them along geometrically distinct directions, and this separation is sharpest at the very layers where the answer is decided. This near-orthogonal routing—not a failure to represent the correct answer—is the structural signature of the knowledge-prediction gap, and it is exactly the misalignment KAPPA is designed to correct.

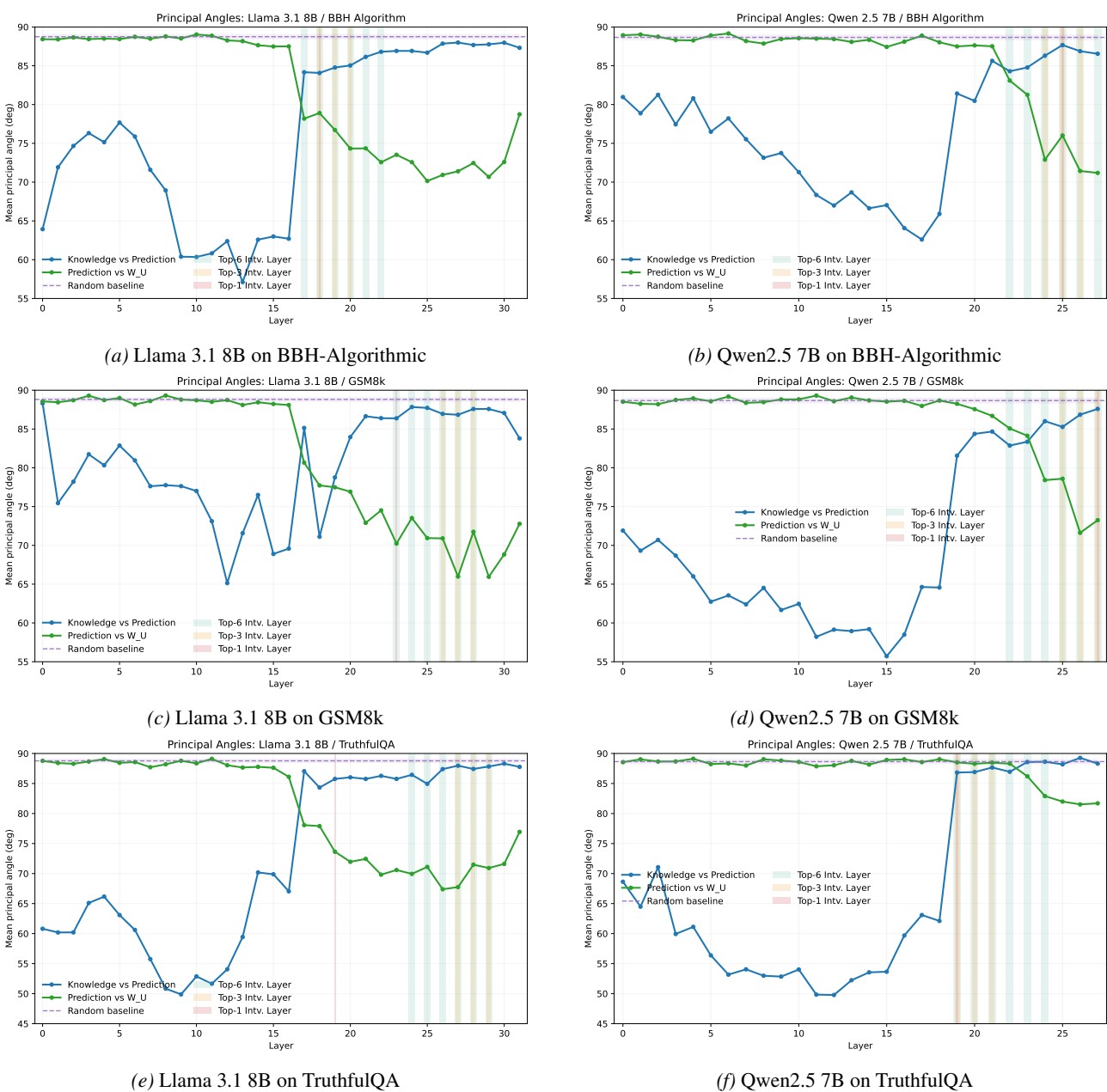

*(a)* Llama 3.1 8B on BBH-Algorithmic

*(b)* Qwen2.5 7B on BBH-Algorithmic

*(c)* Llama 3.1 8B on GSM8k

*(d)* Qwen2.5 7B on GSM8k

*(e)* Llama 3.1 8B on TruthfulQA

*(f)* Qwen2.5 7B on TruthfulQA

*Figure 8.* **Layer-wise mean principal angle analysis across models and benchmarks.** The blue curve shows the mean principal angle between the knowledge and prediction subspaces, while the green curve shows the mean principal angle between the prediction subspace and the corresponding token-specific logit-head subspace from the output unembedding matrix (discussed in §6.1). Vertical lines mark the layers used in our experiments, selected based on high knowledge-probe validation accuracy. The dashed lines indicate the mean principal angles obtained from random subspaces as a reference. In early and middle layers, the knowledge and prediction subspaces are moderately separated, but in the late layers, the mean principal angle rises sharply to nearly orthogonality.

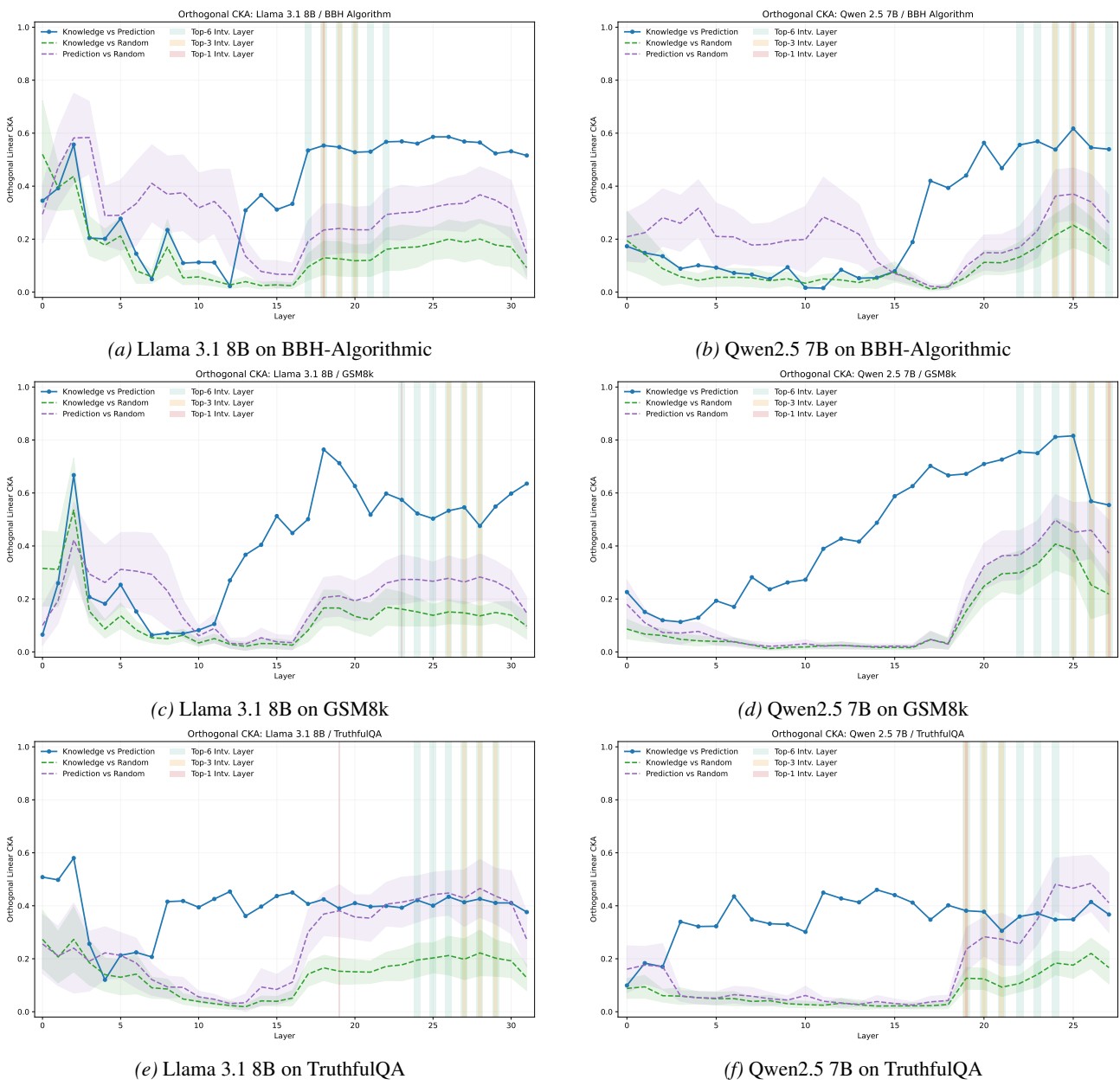

*(a)* Llama 3.1 8B on BBH-Algorithmic

*(b)* Qwen2.5 7B on BBH-Algorithmic

*(c)* Llama 3.1 8B on GSM8k

*(d)* Qwen2.5 7B on GSM8k

*(e)* Llama 3.1 8B on TruthfulQA

*(f)* Qwen2.5 7B on TruthfulQA

*Figure 9.* **Layer-wise orthogonal linear CKA between the knowledge- and prediction-subspace projections across models and benchmarks.** The solid blue curve shows the CKA between representations projected onto the knowledge and prediction subspaces at each layer. The dashed green and purple curves show control CKA values obtained by comparing the knowledge-subspace and prediction-subspace projections, respectively, with matched random subspaces of the same dimensionality. Shaded vertical bands indicate the Top-1, Top-3, and Top-6 KAPPA intervention layers selected in the main experiments. The CKA values remain in an intermediate range (0.4–0.8) at the intervention layers, indicating that the projected representations retain some example-level similarity while still being far from fully aligned at the span level.

## B.5. Connecting the Knowledge-Prediction Gap to Subspace Misalignment

In Appendix B.4, we have shown that knowledge-prediction subspace misalignment exists and becomes most pronounced in the later layers. For this geometric account to be explanatory, however, the degree of misalignment should not merely be present; it should also vary systematically with the magnitude of the knowledge-prediction gap across tasks. We therefore test whether tasks with larger knowledge-prediction gaps also exhibit stronger subspace misalignment.

**Setup.** We extend the analysis to eight benchmarks. For each benchmark, we select the layer at which the validation accuracy of the knowledge probe is highest, and pair the mean principal angle between the knowledge and prediction subspaces at that layer with the knowledge-prediction gap, measured as $1 - \text{AGR}$. We then compute Spearman's rank correlation across benchmarks separately for each model.

**Results.** As shown in Figure 10, the two quantities are strongly related for Llama 3.1 8B: benchmarks with larger subspace misalignment exhibit larger behavioral gaps, with a strong monotonic association (Spearman $\rho = 0.976, p = 0.001$). Qwen2.5 7B shows the same positive trend, though it does not reach significance at this sample size (Spearman $\rho = 0.619$, $p = 0.106$). These correlations provide quantitative support for the geometric account of Section 4: the misalignment between the knowledge and prediction subspaces is not an incidental artifact of the probes but is structurally tied to the behavioral gap that KAPPA targets. The same geometric quantity that distinguishes large-gap from small-gap benchmarks is the one KAPPA acts on, linking the geometry directly to behavior.

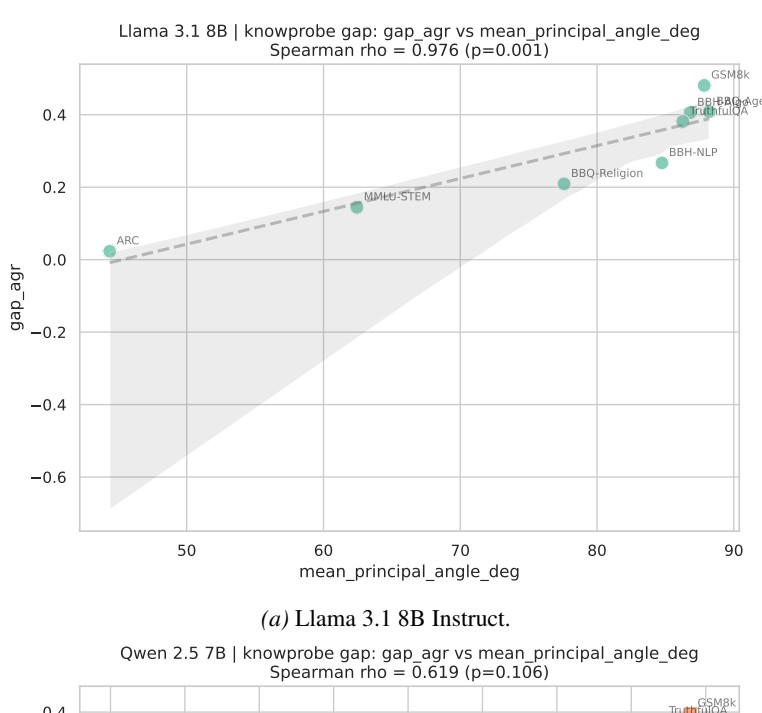

*(a)* Llama 3.1 8B Instruct.

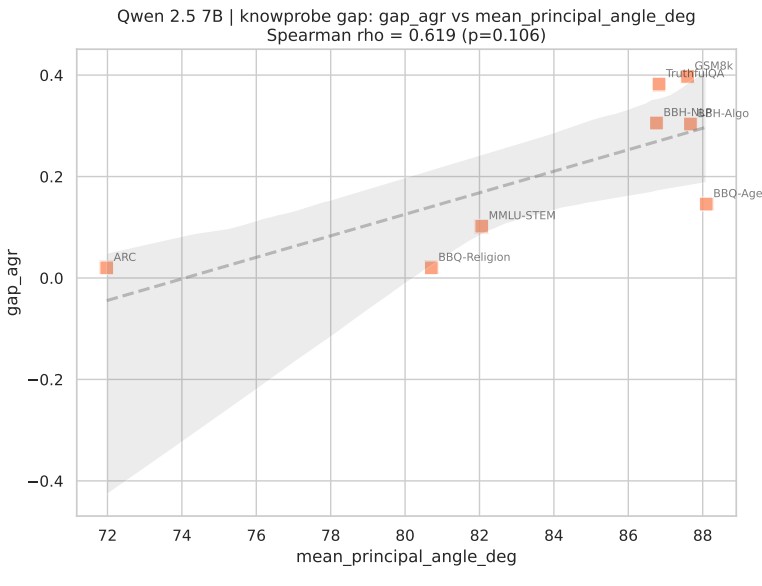

*(b)* Qwen2.5 7B Instruct.

*Figure 10.* **Cross-benchmark relationship between geometric misalignment and the behavioral gap.** Each point corresponds to one benchmark, plotted at the layer achieving the highest validation knowledge-probe accuracy. The $x$-axis is the mean principal angle between the knowledge and prediction subspaces; the $y$-axis is the behavioral gap $1 - \text{AGR}$. Spearman rank correlation $\rho$ is reported in each panel. Benchmarks with larger subspace misalignment exhibit larger behavioral gaps, providing empirical support for our geometric interpretation of the gap.

## B.6. Comparing Prediction Subspace with Logit Space

One possible interpretation is that KAPPA may operate by simply increasing the logit of the option favored by the knowledge probe. If $\mathrm{span}(W_{\mathrm{pred}}^{(l)})$ merely reproduced the token-specific subspace of the unembedding matrix $W_U$, KAPPA would reduce to substituting one set of option-token logits for another. We therefore compare the two subspaces directly.

For each MCQ benchmark we extract the rows of $W_U$ corresponding to its option symbols, forming a $k$-dimensional, token-specific logit-head subspace, and apply the same centering as for the probes (Appendix B.1). At each layer $l$ we measure the mean principal angle between this subspace and $\mathrm{span}(W_{\mathrm{pred}}^{(l)})$.

As the green curve in Figure 8 shows, the angle is close to random in early and middle layers, indicating that the prediction subspace and the logit subspace are largely separated. It decreases substantially in later layers, reaching roughly $65°$–$70°$ at the layers KAPPA selects—well below the random baseline but far from full alignment. The prediction subspace thus drifts toward the output logit space with depth yet remains geometrically distinct even at the intervention layers.

**Implications.** First, because the subspace KAPPA intervenes on is not equivalent to the logit head at that layer, KAPPA does not operate as a token-level logit substitution. Second, the gradual convergence of the prediction subspace toward $W_U$ with depth is consistent with mechanistic-interpretability findings that residual-stream activations encode abstract, output-related features in earlier layers and are progressively unembedded into token-level predictions in later layers (nostalgebraist, 2020; Geva et al., 2023; Lad et al., 2026). Under this view, the prediction subspace is not a copy of the unembedding matrix but a layer-wise prediction geometry that becomes increasingly output-facing with depth. This also explains the transfer behavior of §6: because KAPPA modifies an abstract behavioral subspace rather than a fixed set of token logits, it generalizes to free-form generation (§6.3) and to tasks whose answer symbols or number of options differ from probe training (§6.2).

# C. Derivation of KAPPA

## C.1. Closed-form Update Derivation

In Section 5.1, we solve the following constrained optimization problem:

$$\min_{h'} \ \|h' - h\|_2^2 \qquad \text{s.t.} \qquad \tilde{W}_{\mathrm{pred}}^\top \tilde{h}' = \tilde{W}_{\mathrm{know}}^\top \tilde{h}, \tag{5}$$

given the probe weights $W_{\mathrm{pred}}, W_{\mathrm{know}} \in \mathbb{R}^{d \times k}$ and biases $b_{\mathrm{pred}}$, where we use augmented notation:

$$\tilde{h}' = \begin{bmatrix} h' \\ 1 \end{bmatrix}, \quad \tilde{h} = \begin{bmatrix} h \\ 1 \end{bmatrix},$$

$$\tilde{W}_{\mathrm{know}} = \begin{bmatrix} W_{\mathrm{know}} \\ b_{\mathrm{know}}^\top \end{bmatrix}, \quad \tilde{W}_{\mathrm{pred}} = \begin{bmatrix} W_{\mathrm{pred}} \\ b_{\mathrm{pred}}^\top \end{bmatrix} \in \mathbb{R}^{(d+1) \times k}.$$

While Section 5.1 uses the augmented notation for clarity, here we present a detailed derivation without relying on that notation. To this end, we first rewrite the objective in its non-augmented form:

$$\min_{h'} \ \|h' - h\|_2^2 \quad \text{s.t.} \quad W_{\mathrm{pred}}^\top h' + b_{\mathrm{pred}} = W_{\mathrm{know}}^\top h + b_{\mathrm{know}}.$$

We define the target prediction coordinates

$$p_{\mathrm{target}} := W_{\mathrm{know}}^\top h + b_{\mathrm{know}} - b_{\mathrm{pred}} \in \mathbb{R}^k,$$

so the constraint becomes

$$W_{\mathrm{pred}}^\top h' = p_{\mathrm{target}}.$$

Then we form the Lagrangian

$$\mathcal{L}(h', \lambda) = \|h' - h\|_2^2 + \lambda^\top \left( W_{\mathrm{pred}}^\top h' - p_{\mathrm{target}} \right), \qquad \lambda \in \mathbb{R}^k.$$

Setting the gradient with respect to $h'$ to zero gives

$$2(h' - h) + W_{\mathrm{pred}} \lambda = 0 \quad \Longrightarrow \quad h' = h - \tfrac{1}{2} W_{\mathrm{pred}} \lambda.$$

Substituting into the constraint yields

$$W_{\text{pred}}^\top \left( h - \tfrac{1}{2} W_{\text{pred}} \lambda \right) = p_{\text{target}}.$$

Let

$$G := W_{\text{pred}}^\top W_{\text{pred}} \in \mathbb{R}^{k \times k}$$

denote the Gram matrix. When the columns of $W_{\text{pred}}$ are linearly independent, $G$ is invertible and

$$\lambda = 2G^{-1} \left( W_{\text{pred}}^\top h - p_{\text{target}} \right).$$

Plugging this into the expression for $h'$ gives the closed-form update

$$
\begin{aligned}
h' &= h + W_{\text{pred}} G^{-1} \left( p_{\text{target}} - W_{\text{pred}}^\top h \right) \\
&= h + W_{\text{pred}} \left( W_{\text{pred}}^\top W_{\text{pred}} \right)^{-1} \left( W_{\text{know}}^\top h + b_{\text{know}} - W_{\text{pred}}^\top h - b_{\text{pred}} \right) \\
&= h + W_{\text{pred}} \left( W_{\text{pred}}^\top W_{\text{pred}} \right)^{-1} \left( \widetilde{W}_{\text{know}}^\top \tilde{h} - \widetilde{W}_{\text{pred}}^\top \tilde{h} \right).
\end{aligned}
$$

## C.2. Interpretations of the Update

The solution yields the closed-form update, which is an affine transformation of $h$:

$$h' = \left( I - \frac{W_{\text{pred}} W_{\text{pred}}^\top}{\|W_{\text{pred}}\|^2} \right) h + \frac{p_{\text{target}}}{\|W_{\text{pred}}\|^2} W_{\text{pred}}.$$

Note that the original hidden state $h$ can be decomposed as

$$h = \left( I - \frac{W_{\text{pred}} W_{\text{pred}}^\top}{\|W_{\text{pred}}\|^2} \right) h + \frac{W_{\text{pred}}^\top h}{\|W_{\text{pred}}\|^2} W_{\text{pred}}.$$

Compared to this decomposition, our update $h'$ simply replaces the prediction coordinate $\frac{W_{\text{pred}}^\top h}{\|W_{\text{pred}}\|^2}$ with $\frac{p_{\text{target}}}{\|W_{\text{pred}}\|^2}$, while leaving all orthogonal components unchanged.

## C.3. Extended Alignment

In Section 5.1, we extend our method to capture more diverse knowledge-prediction relationships by introducing hyper-parameters $\alpha, \beta \in \mathbb{R}$. The update with these hyperparameters follows directly from the derivation in Appendix C.1, by substituting the target prediction coordinate with

$$p_{\text{target}} := \alpha \cdot \tilde{W}_{\text{know}}^\top \tilde{h} + \beta \cdot \text{sign}(\tilde{W}_{\text{know}}^\top \tilde{h}) - b_{\text{pred}} \in \mathbb{R}^k.$$

The resulting update is given by

$$
\begin{aligned}
h' &= h + W_{\text{pred}} G^{-1} \left( p_{\text{target}} - W_{\text{pred}}^\top h \right) \\
&= h + W_{\text{pred}} \left( W_{\text{pred}}^\top W_{\text{pred}} \right)^{-1} \left( \alpha \cdot \tilde{W}_{\text{know}}^\top \tilde{h} + \beta \cdot \text{sign}(\tilde{W}_{\text{know}}^\top \tilde{h}) - \widetilde{W}_{\text{pred}}^\top \tilde{h} \right).
\end{aligned}
$$

*Table 11.* Results across knowledge benchmarks for `Qwen2.5 7B Instruct` and `Llama 3.1 8B Instruct`. Higher $\Delta$ACC and AGR indicate smaller knowledge-prediction gaps, while lower KLD reflects closer alignment between encoded knowledge and model predictions. Benchmark names annotated with ($\cdot$) denote the number of answer options in the corresponding MCQ setting. The KP rows report results computed directly from the knowledge probe's predictions. For KAPPA($\cdot$), the value in parentheses denotes the number of layers intervened at inference time. Boldface indicates the best result among {Base, CAA, DoLA, KAPPA($\cdot$)}, excluding the KP row.

| Model | Method | MMLU Humanities | | | MMLU Social Sciences | | | MMLU STEM | | | PubMedQA | | |
|---|---|---|---|---|---|---|---|---|---|---|---|---|---|
| | | ACC | AGR | KLD | ACC | AGR | KLD | ACC | AGR | KLD | ACC | AGR | KLD |
| **Qwen2.5 7B** | Base | 59.91 | 78.50 | 0.777 | 78.85 | 95.93 | 0.718 | 65.17 | 89.70 | **0.703** | 72.33 | 89.80 | 0.570 |
| | CAA | 59.92 | 78.42 | 0.777 | 78.81 | 95.94 | **0.718** | 65.19 | 89.50 | 0.708 | 72.27 | 89.63 | 0.570 |
| | DoLA | 59.85 | 78.64 | 0.811 | 78.52 | 94.81 | 0.725 | **65.60** | 87.81 | 0.714 | 71.63 | 89.63 | 0.570 |
| | KAPPA (1) | 60.77 | 82.03 | 0.751 | 78.91 | **97.12** | 0.734 | 65.24 | **93.17** | 0.721 | 72.57 | **91.43** | **0.565** |
| | KAPPA (3) | 61.04 | 82.23 | 0.720 | 78.95 | 96.78 | 0.739 | 65.45 | 92.70 | 0.724 | 72.53 | 91.37 | 0.565 |
| | KAPPA (6) | **61.92** | **85.22** | **0.703** | **79.12** | 95.47 | 0.759 | 65.56 | 91.99 | 0.764 | **73.00** | 90.70 | 0.588 |
| | KP | 62.47 | 100.00 | 0.000 | 79.02 | 100.00 | 0.000 | 65.36 | 100.00 | 0.000 | 72.00 | 100.00 | 0.000 |
| **Llama 3.1 8B** | Base | 58.57 | 77.94 | 0.469 | **74.04** | 90.55 | 0.471 | 54.65 | **91.12** | **0.321** | 75.73 | 96.43 | **0.434** |
| | CAA | 59.62 | 77.56 | 0.487 | 73.93 | 90.62 | **0.466** | **54.85** | 90.09 | 0.344 | 75.10 | 94.37 | 0.467 |
| | DoLA | 56.08 | 69.35 | 0.504 | 72.25 | 84.00 | 0.571 | 51.76 | 71.36 | 0.533 | 74.63 | 92.53 | 0.536 |
| | KAPPA (1) | 60.67 | **87.15** | 0.533 | 73.79 | **96.08** | 0.617 | 54.34 | 90.56 | 0.684 | 75.17 | 96.77 | 0.576 |
| | KAPPA (3) | 60.32 | 85.19 | 0.456 | 73.24 | 94.54 | 0.607 | 54.03 | 86.30 | 0.755 | 75.33 | **98.57** | 0.566 |
| | KAPPA (6) | **60.73** | 87.00 | **0.446** | 70.49 | 85.98 | 0.778 | 54.23 | 84.58 | 0.899 | **76.03** | 96.97 | 0.599 |

# D. Baseline Details

**Base.** The original large language model (LLM) without any intervention or modification. This baseline reflects the model's raw performance on the evaluation datasets.

**Contrastive Activation Addition (CAA).** Following Rimsky et al. (2024), for each training example, we construct two prompted sequences in the same format, "`user {instruction} question}{option} assistant The answer is (X`", one with the correct option $X = y$ and one with a randomly sampled incorrect option $X \neq y$. We then run the LLM on these sequences and extract the residual stream activation at the final token position from every layer. Let $h^l_{\text{corr}}$ and $h^l_{\text{incorr}}$ denote the resulting activations at layer $l$ for the correct and incorrect sequences, respectively. We define a steering vector at each layer as the mean difference $v^{(l)} = \mathbb{E}\left[h^l_{\text{corr}} - h^l_{\text{incorr}}\right]$.

To select the intervention layer, we apply each candidate vector $v^{(l)}$ once on the validation set by adding it to the residual stream at layer $l$, and choose the single layer that maximizes validation accuracy. We then evaluate on the test set using the selected layer, with a fixed multiplier of $1.0$ for the intervention, following Rimsky et al. (2024).

**Decoding by Contrasting Layers (DoLA).** We evaluate the dynamic variant of DoLA following Chuang et al. (2024). Specifically, we select the optimal layer bucket by evaluating all candidate buckets on the validation set, and then apply the selected bucket for evaluation on the test set. Since our setting focuses on single-token prediction, we set the hyperparameter $\alpha = 0$ throughout the experiments.

# E. Additional Results

## E.1. KAPPA on MCQ Benchmarks

**Results Across Datasets.** Table 11 reports results on knowledge-oriented benchmarks for `Qwen2.5 7B Instruct` and `Llama 3.1 8B Instruct`. Across these datasets, where the knowledge-prediction gap is relatively smaller than in reasoning or bias benchmarks, we still observe that KAPPA consistently improves agreement (AGR), indicating more faithful alignment between linearly accessible knowledge and model predictions. At the same time, gains in task accuracy (ACC) remain modest. This is expected, as the performance of the knowledge probe itself is comparatively low on these benchmarks, limiting the potential upper bound for accuracy improvements. Nevertheless, KAPPA yields non-trivial accuracy gains in some settings, suggesting that even when the gap is less pronounced, improving representation-level alignment can still translate into more reliable predictions.

**Results Across Models.** Table 12 reports the average accuracy (ACC), agreement rate (AGR), and KL divergence (KLD) across multiple models on the GSM8k benchmark. Across all model families, KAPPA consistently increases AGR relative to the base model, demonstrating its effectiveness in mitigating the knowledge-prediction gap. This suggests that KAPPA reliably encourages models to better utilize their linearly accessible knowledge during inference.

*Table 12.* Average accuracy (ACC), agreement (AGR), and KL divergence (KLD) across multiple models on the GSM8k benchmark. For KAPPA(·), the value in parentheses denotes the number of layers intervened at inference time. Boldface indicates the best result among {Base, CAA, DoLA, KAPPA(·)}, excluding the knowledge probe (KP).

| Method | GSM8k | | | | | | | | | | | | | | |
|---|---|---|---|---|---|---|---|---|---|---|---|---|---|---|---|
| | Mistral v0.3 7B | | | Llama-3.1 8B | | | Qwen2.5 7B | | | Qwen3 4B | | | Qwen3 14B | | |
| | ACC | AGR | KLD | ACC | AGR | KLD | ACC | AGR | KLD | ACC | AGR | KLD | ACC | AGR | KLD |
| Base | 31.7 | 58.3 | 0.41 | 32.6 | 53.7 | 0.35 | 47.8 | 60.3 | 0.86 | 47.1 | 70.4 | 0.57 | 63.5 | 63.5 | 0.74 |
| CAA | 31.8 | 58.4 | 0.42 | 32.9 | 53.8 | 0.38 | 47.7 | 60.3 | 0.85 | 47.1 | 70.5 | 0.62 | **64.3** | 63.1 | 0.76 |
| DoLA | 31.7 | 55.7 | 0.62 | 33.2 | 49.7 | 0.58 | 48.1 | 60.0 | 0.86 | 46.0 | 66.1 | 0.67 | 62.6 | 60.7 | 0.77 |
| KAPPA (1) | 33.0 | 79.0 | 0.54 | 34.9 | **73.8** | 0.37 | 49.6 | 68.8 | 0.78 | **47.5** | **76.2** | **0.56** | 53.0 | **73.8** | 0.74 |
| KAPPA (3) | 33.6 | **83.8** | 0.48 | **34.6** | 66.2 | 0.27 | 49.1 | 65.4 | 0.77 | 47.2 | 72.8 | 0.71 | 61.3 | 72.7 | **0.72** |
| KAPPA (6) | **33.6** | 72.5 | 0.49 | 36.6 | 75.9 | **0.27** | **49.2** | **66.3** | **0.76** | 46.7 | 73.2 | 0.70 | 60.8 | 69.0 | 0.79 |
| KP | 34.0 | 100.0 | 0.00 | 36.6 | 100.0 | 0.00 | 50.3 | 100.0 | 0.00 | 46.7 | 100.0 | 0.00 | 56.6 | 100.0 | 0.00 |

*Table 13.* **Results on the TruthfulQA benchmark across multiple models.** All models use their instruct variants. BASE denotes the original model without intervention, KP denotes the knowledge probe accuracy, and KAPPA denotes our single-layer intervention. All configurations are identical to those used in the original manuscript, and we report results for two values of $\alpha$, 30.0 and 80.0. Boldface marks the best result among BASE and the two KAPPA settings. Higher is better for ACC and AGR, and lower is better for KLD.

| | Mistral v0.3 7B | | | Llama-3.1 8B | | | Qwen2.5 7B | | |
|---|---|---|---|---|---|---|---|---|---|
| | ACC | AGR | KLD | ACC | AGR | KLD | ACC | AGR | KLD |
| Base | 40.7 | 46.6 | 0.936 | 56.7 | 62.1 | 0.629 | 58.8 | 61.8 | 1.006 |
| KAPPA ($\alpha$=30) | 51.0 | 59.7 | 0.820 | 67.8 | 78.8 | **0.599** | 60.6 | 64.0 | 0.992 |
| KAPPA ($\alpha$=80) | **58.7** | **71.3** | **0.799** | **70.3** | **82.6** | 0.676 | **64.2** | **68.3** | **0.952** |
| KP | 69.5 | 100.0 | 0.000 | 76.3 | 100.0 | 0.000 | 80.1 | 100.0 | 0.000 |

| | Qwen3 4B | | | Qwen3 14B | | | Qwen3 32B | | |
|---|---|---|---|---|---|---|---|---|---|
| | ACC | AGR | KLD | ACC | AGR | KLD | ACC | AGR | KLD |
| Base | 56.5 | 60.0 | 0.820 | 71.6 | 76.0 | 0.762 | 79.5 | 82.2 | **0.642** |
| KAPPA ($\alpha$=30) | 58.4 | 62.3 | 0.795 | 73.3 | 78.1 | 0.759 | 81.3 | 84.7 | 0.644 |
| KAPPA ($\alpha$=80) | **61.4** | **66.3** | **0.775** | **75.5** | **81.3** | **0.755** | **84.0** | **88.2** | 0.643 |
| KP | 78.7 | 100.0 | 0.000 | 85.8 | 100.0 | 0.000 | 89.1 | 100.0 | 0.000 |

## E.2. Effect of Model Scale

We further examine how the knowledge-prediction gap and the effectiveness of KAPPA vary with model scale. Table 13 reports results on TruthfulQA across models of different sizes, including an extended analysis with a larger model, `Qwen3-32B`. While base prediction accuracy improves with scale, a non-trivial gap between base prediction and knowledge probe accuracy remains even for `Qwen3-32B` (9.6%). KAPPA consistently improves accuracy and reduces this gap across all model sizes, suggesting that larger models also underutilize internally encoded knowledge.

**Discussion.** We further speculate on potential causes of the knowledge-prediction gap: model *scale* and *distillation*. Smaller models may lack the capability to reliably translate internally encoded knowledge into correct predictions, leading to a larger gap. Furthermore, when trained via distillation, smaller models might capture the rich knowledge transferred from larger models within their intermediate representations, yet still lack the capability to utilize it reliably.

Our results provide partial evidence for disentangling the roles of scale and distillation. Comparing `Qwen3-4B` and `Qwen3-32B`, the difference in knowledge probe accuracy is relatively modest (∼10% points), whereas the difference in base prediction accuracy is much larger (∼23% points). This suggests that distillation may transfer a substantial portion of knowledge to smaller models, but that these models still struggle to translate such knowledge into correct predictions. However, the gap is not solely attributable to distillation: non-distilled models such as `Mistral-v0.3-7B-Instruct` and `Llama-3.1-8B-Instruct` also exhibit substantial gaps (28.8% and 19.6%, respectively). Finally, although the gap generally decreases with scale, it does not disappear, indicating that scaling alone is insufficient to fully resolve the misalignment between internal knowledge and model predictions. Overall, these findings suggest that the knowledge-prediction gap arises from multiple factors, including model scale and training dynamics such as distillation.

*Table 14.* **Robustness to distractor sampling.** We construct 10 test sets using different random seeds for distractor sampling (sampling only the test sets) and report the mean and standard deviation of accuracy across these runs for Qwen2.5 7B Instruct and Llama-3.1 8B Instruct. All experimental configurations (e.g., layer selection and hyperparameters) are kept identical to those in the main tables. KAPPA (1) and KAPPA (6) denote applying the intervention to a single layer and six layers, respectively.

| Model | Method | BBH-algo | | TruthfulQA | |
|---|---|---|---|---|---|
| | | Mean | Std. Dev. | Mean | Std. Dev. |
| Qwen2.5 7B Instruct | Base | 51.67 | 1.281 | 59.93 | 1.577 |
| | KAPPA (1) | 51.90 | 1.428 | 61.74 | 1.558 |
| | KAPPA (6) | 53.85 | 1.850 | 64.71 | 1.558 |
| Llama 3.1 8B Instruct | Base | 44.55 | 1.303 | 56.38 | 1.530 |
| | KAPPA (1) | 48.95 | 1.390 | 67.36 | 2.053 |
| | KAPPA (6) | 49.20 | 1.842 | 73.73 | 2.529 |

*Table 15.* **Statistical significance under distractor sampling.** We construct 10 test sets using different random seeds for distractor sampling (sampling only the test sets) and perform one-sided t-tests on accuracy to evaluate whether KAPPA outperforms the base model. Results are reported for Qwen2.5 7B Instruct and Llama-3.1 8B Instruct, using the same experimental configurations as in the main tables. KAPPA (1) and KAPPA (6) denote applying the intervention to a single layer and six layers, respectively.

| Model | Dataset | KAPPA (1) p-value | KAPPA (6) p-value |
|---|---|---|---|
| Qwen2.5 7B Instruct | BBH-algo | 0.105 | $< 0.00001$ |
| Qwen2.5 7B Instruct | TruthfulQA | $< 0.00002$ | $< 0.00000001$ |
| Llama 3.1 8B Instruct | BBH-algo | $< 0.000001$ | $< 0.0001$ |
| Llama 3.1 8B Instruct | TruthfulQA | $< 0.00000001$ | $< 0.00001$ |

### E.3. Sensitivity Analysis

**Sensitivity to distractor sampling.** Because BBH and TruthfulQA contain varying numbers of answer choices across instances, we construct four-choice evaluation sets by randomly sampling distractors. To assess the sensitivity of our results to this sampling process, we repeat the KAPPA intervention experiments using 10 different random seeds for distractor sampling while keeping the learned subspaces and all other experimental configurations fixed. As shown in Table 14, KAPPA consistently improves mean accuracy across all settings for both Qwen2.5 7B Instruct and Llama-3.1 8B Instruct, with relatively small standard deviations (1.281–2.529), indicating that the gains are stable across different distractor sets. Furthermore, Table 15 shows that the improvements are statistically significant in most settings, with near-zero p-values under one-sided t-tests against the base model. These results suggest that KAPPA is robust to variations in distractor sampling and that the reported improvements are not artifacts of a particular distractor configuration.

**Sensitivity to probe training data size.** To assess sensitivity to probe training data size, we subsample the training data at 10, 20, 40, 60, and 80%, retrain the probes on each subset using three random seeds, and apply KAPPA using the resulting subspaces. As shown in Figure 11, KAPPA consistently improves over the base model even when trained with only 10% of the data, while performance becomes more stable as the dataset size increases. Figure 12 further shows that knowledge probe accuracy follows a trend similar to KAPPA performance, suggesting that KAPPA can effectively leverage knowledge signals even in low-data regimes. Figure 13 shows that prediction probe accuracy remains consistently high (around or above 90%) even with small training subsets, indicating relatively low sensitivity to training data size. Overall, these results suggest that KAPPA remains effective and practically applicable even when only limited probe training data is available.

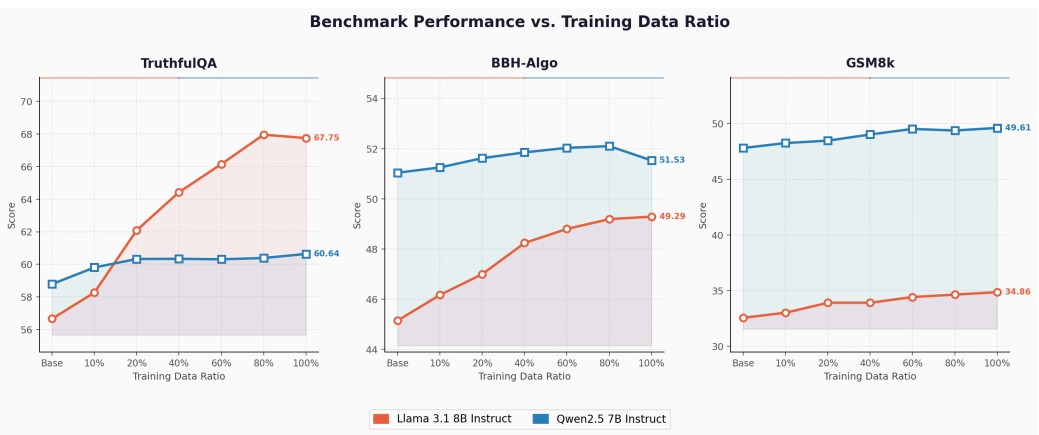

*Figure 11.* **KAPPA accuracy under varying training data sizes.** We subsample the training data at 10, 20, 40, 60, and 80% using three random seeds, learn subspaces from each subset, and report the resulting KAPPA accuracy (averaged over seeds). Results for Llama-3.1 8B Instruct and Qwen2.5 7B Instruct are shown with different colored lines. All configurations (e.g., layer selection and hyperparameters) follow KAPPA (1-layer) from the main table.

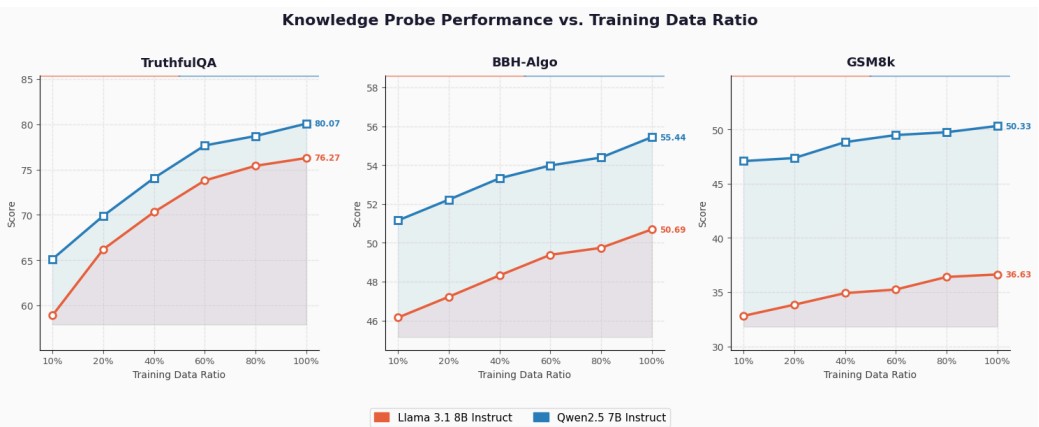

*Figure 12.* **Knowledge probe accuracy under varying training data sizes.** We train knowledge probes on 10, 20, 40, 60, and 80% subsampled datasets (three random seeds) and report accuracy averaged over seeds. Results for Llama-3.1 8B Instruct and Qwen2.5 7B Instruct are shown with different colored lines. All configurations follow KAPPA (1-layer) from the main table.

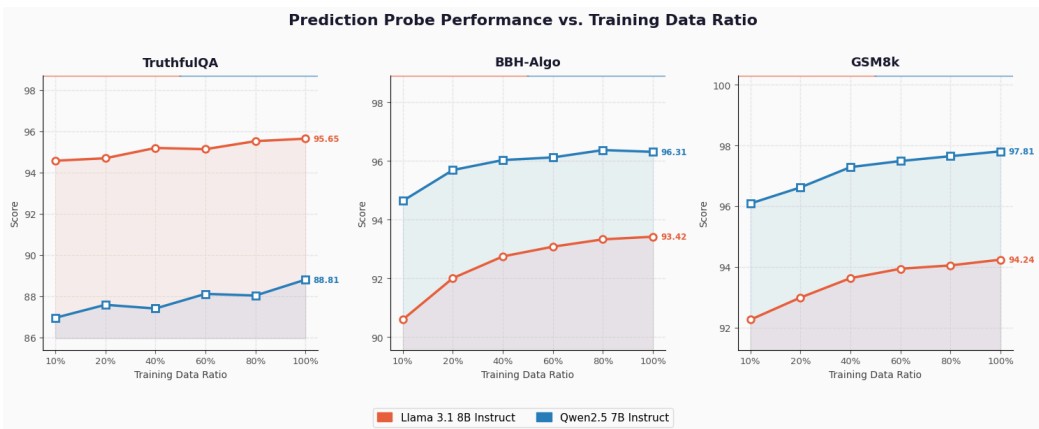

*Figure 13.* **Prediction probe accuracy under varying training data sizes.** We train prediction probes on 10, 20, 40, 60, and 80% subsampled datasets (three random seeds) and report accuracy averaged over seeds. Results for Llama-3.1 8B Instruct and Qwen2.5 7B Instruct are shown with different colored lines. All configurations follow KAPPA (1-layer) from the main table.

## E.4. Extended Results for Cross-Dataset Transfer

Figure 14 extends Figure 6 by adding an additional panel that analyzes cases where KAPPA flips originally correct predictions into incorrect ones, providing a more complete and fine-grained view of its transfer behavior. Overall, the results are largely consistent with the trends observed in our original gain-only analysis, while revealing additional differences across source-target subspace pairs. In particular, when subspaces extracted from GSM8k or BBH-NLP are transferred to BBH-Algorithmic, the gains are generally small whereas the losses are relatively large, especially on subsets such as *Temporal Sequences*. This suggests that algorithmic tasks rely on a more distinct subspace that does not transfer well from other domains. In contrast, when subspaces from GSM8k or BBH-Algorithmic are transferred to BBH-NLP, reasoning-oriented subsets such as *Date Understanding* and *Penguins in a Table* exhibit larger gains and smaller losses, whereas other NLP-related subsets such as *Movie Recommendation* and *Ruin Names* show the opposite pattern.

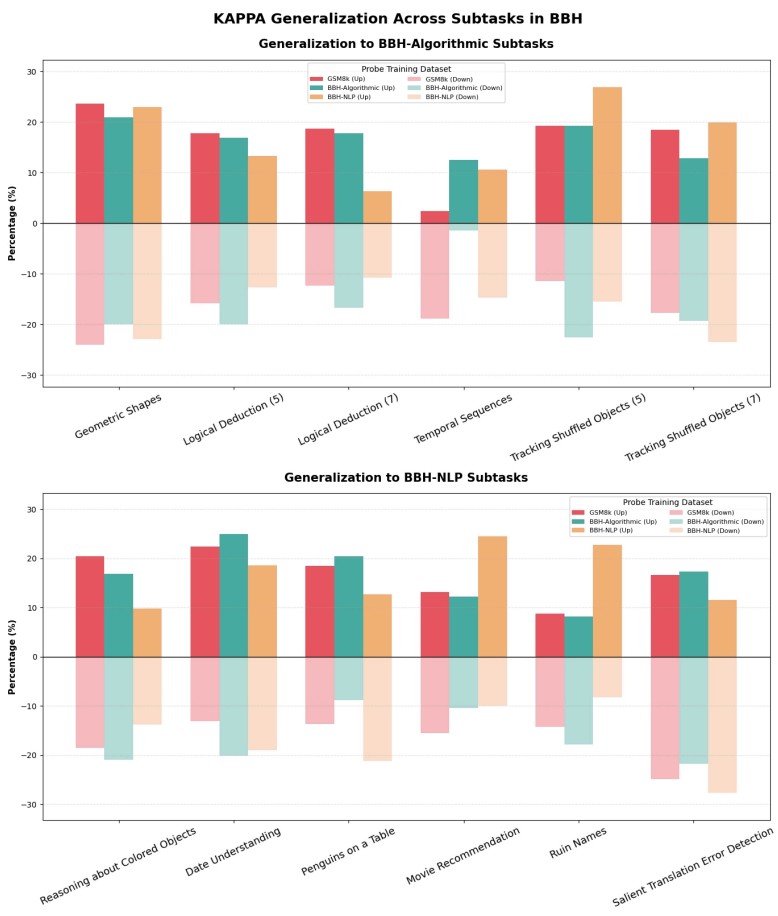

*Figure 14.* **Per-subtask analysis of KAPPA generalization on BBH (Llama-3.1 8B Instruct).** For each probe training dataset (GSM8k, BBH-Algorithmic, BBH-NLP), we identify both incorrect-to-correct and correct-to-incorrect transitions after applying KAPPA, and report their distribution across BBH subtasks. Upward bars indicate gains (incorrect-to-correct transitions), while downward bars indicate losses (correct-to-incorrect transitions).

