# OpenReview forum: "Bridging the Knowledge-Prediction Gap in LLMs on Multiple-Choice Questions"
_ICML.cc/2026/Conference — ICML 2026 regular_

### Official Review · Reviewer_Dryz · 2026-02-23

**Soundness:** 3
**Presentation:** 3
**Significance:** 2
**Originality:** 2
**Overall Recommendation:** 4
**Confidence:** 3

**Summary:**

This paper concerns settings where an LLM has the capacity to answer a question correctly (ie. that “knowledge” exists in its representations), but fails to do so. The authors show that at a given layer, the same hidden state simultaneously contains (1) a linearly decodable signal for the correct answer and (2) a linearly decodable signal for the model’s eventual (possibly wrong) choice. Those two signals are not aligned. The paper authors quantify this disconnect through the knowledge-prediction gap, apply a mechanistic interpretability approach to identify the contributing subspaces, and introduce an inference-time intervention to minimize the gap. They find that the knowledge probe significantly outperforms generation accuracy; the prediction probe outperforms the knowledge probe on predicting its own choice. They introduce KAPPA, a technique used to close the gap between knowledge and prediction.

**Compliance With Llm Reviewing Policy:**

Affirmed.

**Final Justification:**

Concerns addressed; score bumped

**Key Questions For Authors:**

1. The prediction subspace could simply approximate the model’s existing logit head. Can the observed “misalignment” reduce to probe-vs-head differences rather than a deeper representational phenomenon?
2. Can the knowledge-prediction gap be generalized to be used as an uncertainty metric?
3. Is 'knowledge' the right word to describe your object of study?

**Limitations:**

yes

**Strengths And Weaknesses:**

**Soundness:**

A few parts of the paper were particularly well-done.

- It was good to distinguish between different types of knowledge and to clarify that your approach only concerns knowledge that it accessible through linear operations on hidden states.
- The KAPPA results were strong.
- Many different model families, benchmarks, and performance metrics were used. These constitute robustness checks.

The paper has a few areas with soundness concerns.

- While a prediction probe is necessary to construct a linear prediction subspace for intervention, it remains unclear whether the learned prediction subspace captures anything beyond a linear approximation to the model’s logit head. How does the prediction probe differ from directly analyzing the model’s own output logits?
- I found the framing of your linear probe logits as “coordinates in a k-dimensional subspace of the model’s representation” confusing. These values are functions of the underlying representations, and as such, live in a fundamentally different space than the representation itself. It would help to clarify in what sense these logits can be interpreted as coordinates of the representation itself, rather than coordinates in a probe-defined feature space.
- The paper equates “linearly decodable from hidden states” with “the model possesses knowledge.” Linear separability does not imply the information is causally used in generation, and may reflect spurious correlations.
- The experiments are thorough and technically solid, but the causal interpretation (linear decodability = knowledge; geometric misalignment as explanatory mechanism) is stronger than the evidence strictly warrants.

**Nits (not used to determine final scores)**

- This sentence was confusing to me: “Furthermore, since this is the easiest form of knowledge to exploit, its common underutilization makes it implausible that the model effectively leverages more complex forms of knowledge.” Can you expand on this idea?


**Presentation**:

The submission is mostly clearly written and well structured. I would suggest the following:

- In Figure 1, KAPPA is illustrated but not described. For your eventual readers, it would be helpful to include a pointer to your KAPPA method in the figure caption.
- Figure 5 is unclear to me. It would help to explicitly state in the caption that positive values indicate agreement improvement over base and negative values indicate degradation.
- In Figure 6, it would be interesting to have a panel that looks at the same information for questions where KAPPA was made an initially-correct answer incorrect.


**Significance***:*

LLMs have a documented reliability problem; they are sensitive to prompts, history effects, ordering effects, etc. This paper surely addresses an important and relevant problem because it tries to quantify one aspect of this unreliability: the knowledge-prediction gap.

**Originality**:

The mechanism of extracting a correct answer from a hidden representation is a technique that already existed in the mechanistic interpretability literature. The authors extend on this line of work by providing a method to close the gap. The identification of the knowledge-prediction gap is unique, as is the approach to closing the gap. Linear probing, activation steering, and projection are all existing techniques, but their combination in relation to the KPG is new.

---

> ### Author Rebuttal · Authors · 2026-03-31
>
> We thank the reviewer for recognizing the strength and robustness of our results, the importance of the problem, and the uniqueness of our approach.
>
> ## W1. Unclear distinction between prediction subspace and logit head
>
> The prediction probe is conceptually distinct from the model's logit head: it is a **layer-specific linear readout** trained to predict the model's MCQ choice from residual-stream activations, rather than relying on the fixed unembedding matrix $W_U$. We do not expect the two to remain fully separate across depth—as the model’s final decision becomes progressively formed, the prediction subspace should become increasingly aligned with the output geometry of $W_U$.
>
> https://anonymous.4open.science/r/KAPPA-Rebuttal-66D2/R4-1.pdf
>
> To test this, we compared the prediction subspace with $W_U$ across layers using principal angles. In early and middle layers, the angle stays close to the random-subspace baseline, while in later layers it decreases substantially (reaching around 65°–70°), indicating partial alignment but not equivalence. We therefore interpret the prediction probe as capturing **a layerwise prediction geometry that increasingly reflects output geometry over depth.** This distinction is important: **KAPPA intervenes at layers where prediction-related structure is already linearly accessible but not yet identical to the final logit-space representation.**
>
> ## W2. Confusing interpretation of probe logits as representation coordinates
> We thank the reviewer for this helpful clarification and agree that the distinction was not sufficiently clear.
>
> Our intended interpretation is as follows: the columns of the probe weight matrix define a $k$-dimensional subspace in $\mathbb{R}^d$, and $W^\top h$ corresponds to projecting $h$ onto this subspace. Thus, **the probe logits represent the coordinates of $h$ within the probe-defined subspace, rather than the full representation.**
>
> We will clarify this in Section~3.2 by revising the phrasing to: “**the projection of $h$ onto the subspace spanned by the probe weight vectors, representing its position within this probe-defined subspace.**”
>
> ## W3. Overstated causal interpretation of linear decodability and geometric explanation
>
> It was not our intention to equate knowledge possession/existence (linear decodability) with knowledge realization. Since we acknowledge that **existing/decodable knowledge and actual predictions can diverge,** we wanted to investigate the reasons behind it. While the geometric misalignment does not guarantee causal relationships by design, our invention experiments clearly show the causal effects of subspace alignment, reducing the gap (higher AGR, lower KLD), **indicating that the knowledge signal can influence predictions when properly integrated**. In the revision, we will clarify this distinction and avoid phrasing that equates linear decodability with causality.
>
> ## W4. Clarification for the claim about the underutilized simple knowledge
>
> We agree that this sentence may be confusing and does not clearly convey our intended point. We intended to highlight that **even linearly accessible information is not fully reflected in the model’s final predictions.** In the revised manuscript, we will either soften this statement or remove it to avoid confusion.
>
> ## W5. Suggestions for figures and captions
>
> We thank the reviewer for the helpful suggestions.
>
> - **Figure 1.** We will revise the caption to clearly reference the KAPPA method (Section 3).
> - **Figure 5.** We will clarify that positive values indicate improved agreement over the base model, while negative values indicate degradation.
> - **Figure 6.** We will add a panel analyzing cases where KAPPA flips correct predictions to incorrect ones, providing a more complete and fine-grained view. (https://anonymous.4open.science/r/KAPPA-Rebuttal-66D2/R4-2.pdf)
>
> ## Q1. Is the prediction subspace meaningfully distinct from the logit head?
> See response to W1.
>
> ## Q2. Can the knowledge-prediction gap be generalized to be used as an uncertainty metric?
>
> We see this as a promising direction, but do not establish the gap as a calibrated uncertainty measure and caution against such an interpretation. Unlike standard uncertainty measures (e.g., entropy, softmax confidence) defined over the output distribution, **the knowledge-prediction gap operates in representation space.** We interpret it as a signal of internal inconsistency—a mismatch between internal knowledge and final predictions. If uncertainty is framed as confidence over correctness, the knowledge probe logit values may serve as a **proxy for the model’s internal belief.**
>
> ## Q3. Is 'knowledge' the right word to describe your object of study?
>
> We use *knowledge* to **refer to signals in the model’s representations that are predictive of the correct answer.** We acknowledge that this term may be imperfect and are open to adopting a more precise alternative if the reviewer has a suggestion.

---

> > ### Author Rebuttal · Reviewer_Dryz · 2026-04-01
> >
> > Thanks for the thoughtful rebuttal. I have bumped up my score one point in response.

---

### Official Review · Reviewer_9GNF · 2026-03-08

**Soundness:** 2
**Presentation:** 1
**Significance:** 3
**Originality:** 3
**Overall Recommendation:** 4
**Confidence:** 3

**Summary:**

This paper presents KAPPA, an inference-time steering mechanism that aims to improve multiple-choice answering by correcting a model's residual stream activations, steering them toward an internally encoded "knowledge" signal. The authors first quantify a knowledge-prediction gap, the discrepancy between what linear probes can extract from hidden states and what the model actually outputs, across diverse MCQ benchmarks. They provide a geometric interpretation via misaligned knowledge and prediction subspaces, and propose KAPPA, which applies a minimal affine correction to align the two. Experiments across several models (4B-14B) and benchmarks show consistent improvements, particularly on truthfulness and bias tasks.

**Compliance With Llm Reviewing Policy:**

Affirmed.

**Final Justification:**

The paper evaluates an interesting question of leveraging "internal knowledge" to improve performance at inference time. The empirical evaluation is broad, and the method seems to be effective. The authors made a good effort to rebut most of the points raised in the review, leaving minor concerns around the justification/expressiveness of generalized alignment. Therefore, I am leaning towards accept.

**Key Questions For Authors:**

1. Why is linear scaling of the projection necessary through alpha? Is there a theoretical reasoning behind the choice of a single-parameter scaling?

2.  Regarding the prediction probe accuracy, it seems unfeasible to me, given the randomness that may occur during decoding, to attain close to 100% accuracy in predicting the decoded token from the later hidden layer representations on an evaluation set. Beyond, to my reading, the “prediction dataset” was generated using a single model generation - would results become more consistent if you accounted for a larger sample of the predictive distribution?

3. How sensitive is KAPPA to the probe training data size? The current setup uses relatively large training sets (Table 7). Would performance degrade substantially with fewer labeled examples?

**Limitations:**

yes

**Strengths And Weaknesses:**

Strengths:

- The paper interestingly proposes leveraging the benefits of "internal knowledge" to improve performance at inference time. The approach is general enough to apply to any MCQ-answering task.

- Empirical work is rather thorough, with evaluations across many datasets, models, and ablations including cross-dataset transfer and free-form generation.


Weaknesses / Suggestions:

- I am unsure about the theoretical grounding of the main method discussed in §4.1. Linear alignment between representation spaces is thoroughly discussed in representational alignment literature (CCA/CKA; …), but the connection to it is unclear and missing from related work. For instance, linear alignment usually works with approximate measures of equality, as exact equality is often too restrictive; I wonder whether the method would benefit the method from more permissive constraints.

- Similarly, the parametrized generalization of the method (“Generalized Alignment”) seems unintuitive/lacking a formal motivation, as well as not very expressive. To my reading, $\alpha$ is a uniform positive scalar applied to all logits, and the $\beta$ term essentially gives you pointwise (positive or negative) sharpening of that distribution of the same magnitude across $k$.

- Given the size of the training data (cf. Tab.7), I’d be interested in a comparison to using internal knowledge for retraining models and how that differs from these steering interventions.

- The authors should be careful about their reporting. For instance, from looking at the plots in Fig 2 and Appendix D.1, the knowledge probe accuracy in fact does not seem to exceed 90%, whereas the prediction probes are close to 100% overall. This is inconsistent with what is written in L.189-204 (second column). Another example is the claim that the prediction gap can help reduce hallucinations made in L.234. More evidence is needed beyond a single model evaluation on one dataset to support such a claim. Similarly, Figure 3 nicely shows stronger linear correlation on some datasets. Instead of relying on visual intuition to support your claim, it would be more convincing to simply compute the correlation coefficient.

- Some of the notation/presentation could be substantially improved, for instance, in §3.1, the prediction agreement and KLD are defined as a function of $x$, but $x$ does not appear in the actual term.  I suggest notationally highlighting the implicit dependence of $p_K$ and $p_M$ on the instance $x$. Beyond, both metrics are actually reported averaged across the eval set in all tables. I suspect it’s more easily understandable if you explicitly write this. The tilde notation introduced in Eq. 3 is dropped in Appendix B.1, and no derivation of the closed-form solution is neither derived nor shown for the “Generalized Alignment” presented in Eq. 4.

- It is worth noting that KAPPA's perturbation is constrained to the column space of $W_{\mathrm pred}$, which is at most k-dimensional ($k \in  \{3,4\}$ for the benchmarks considered), while the residual stream lives in $R^d$ with d on the order of thousands. I suspect that means the intervention operates in an extremely low-rank subspace. While the authors frame this as a feature (minimal perturbation), it also raises the question of whether such a narrow correction can capture the full extent of the misalignment. If the knowledge-prediction gap has components outside this tiny subspace, then KAPPA is inherently limited by construction. Some analysis of the residual error after projection, or exploration of whether a slightly higher-dimensional intervention space could close more of the gap, would strengthen the paper.

Minor:
- I suspect the bolding in Table 1 is incorrect; AGR should be bolded according to larger values being better, and the KLD column should bold the smallest values. This seems to be correctly done in all other tables.
- L.203: by convention, the probability distribution over $k$ elements lies in the $(k−1)$-dimensional probability simplex.

---

> ### Author Rebuttal · Authors · 2026-03-31
>
> We thank the reviewer for the encouraging feedback. We appreciate the recognition that our approach is interesting, broadly applicable, and supported by thorough empirical evaluation.
>
> ## W1. Unclear connection to representation alignment and the strictness of exact equality constraints
>
> We agree that the original draft did not sufficiently situate KAPPA within the representational alignment literature and will revise the related work accordingly. While CCA/CKA quantify similarity between representation spaces, **KAPPA addresses an intervention problem.**
>
> https://anonymous.4open.science/r/KAPPA-Rebuttal-66D2/R3-1.pdf
>
> To better ground this connection, we quantify the relation between the knowledge and prediction subspaces using mean principal angles and orthogonal linear CKA. The results show that **the two subspaces are largely separated at the span level, while retaining partial structural similarity.** We will include these in the revision and discuss them in relation to the representational alignment literature (see ecLd-W1 for more details).
>
> We appreciate the suggestion on soft alignment. Our use of a hard equality constraint follows directly from the formulation, which **aligns prediction logits with knowledge logits under a minimum $\ell_2$ perturbation objective, yielding a closed-form solution.** We will clarify this and highlight soft alignment as future work.
> ## W2. Limited justifications of generalized alignment
> Eq. 4 is not intended as a rigid objective, but as a **simple parameterization for controlling alignment strength**: $\alpha$ rescales the alignment, and $\beta$ adds a sign-based offset that enforces a margin toward the knowledge-preferred direction. As shown in Section 4.2, these parameters **directly control the degree of alignment**, serving as simple, tunable controls. More expressive parameterizations are possible, and we will clarify this in the revision.
> ## W3. Missing comparison with retraining-based approaches
> KAPPA operates as an **inference-time intervention** and is therefore fundamentally different from retraining-based approaches. While retraining updates model weights, **KAPPA keeps them fixed and directly modifies residual stream activations.** It operates at a single (or a few) layer(s) with far fewer parameters, making it lightweight and efficient. **As such, KAPPA and retraining are complementary rather than competing approaches.**
> ## W4-W5-W7. Issues in presentation, notation, and reporting
> Thank you for the detailed feedback on the presentation. We take this seriously and will incorporate all suggested revisions in the final manuscript, including:
> - Correcting inconsistency between Fig. 2 / Appendix D.1 and L.189–204
> - Refining the hallucination claim in L.234
> - Adding correlations to Fig. 3
> - Clarifying definitions of AGR/KLD
> - Ensuring use of tilde notation for Eq.3 in Appendix B.1
> - Providing derivations for Eq.4 (generalized alignment)
> - Fixing Table 1 bolding criteria
> - Correcting the description of the probability simplex
> ## W6. Potential limitation due to low-rank intervention space
> We agree that KAPPA is by construction a low-dimensional intervention: the closed-form update lies in the $k$-dimensional column space of $W_{\text{pred}}$. This is **not an artifact of approximation, but a direct consequence of our objective**: to correct the probe-defined coordinate mismatch by aligning $W_{\text{pred}}^\top h'$ with $W_{\text{know}}^\top h$. Under this formulation, the resulting update is an effective way to **exactly achieve the objective while incurring minimal perturbation.**
> We also acknowledge that additional misalignment may exist beyond this subspace; extending the formulation to higher-dimensional interventions is a promising direction for future work
> ## Q1. Justification for the single-parameter scaling
> Please refer to W2.
> ## Q2. Reliability of the prediction probe under decoding randomness
> Our prediction dataset uses **greedy decoding rather than stochastic sampling.** This aligns with our goal of matching the model’s internal confidence about the correctness of each option with its predictive confidence. By taking the highest-probability option as the prediction, we obtain a target **reflecting the model’s most confident decision**, enabling us to identify a subspace that effectively captures predictive confidence. We will clarify this in the revision.
> ## Q3. Sensitivity to probe training data size
> We have evaluated sensitivity to probe training data by sampling the training set to 10–80\% (three random seeds), retraining probes, and applying KAPPA. While larger datasets improve performance, **KAPPA consistently outperforms the base model even with 10% of the data**, with prediction probe accuracy remaining around 90% or higher. This suggests that **KAPPA remains effective in low-data regimes**, supporting its practical applicability. We will include this analysis in the revision.
> https://anonymous.4open.science/r/KAPPA-Rebuttal-66D2/R3-2.pdf

---

> > ### Author Rebuttal · Reviewer_9GNF · 2026-04-03
> >
> > I appreciate the authors' efforts to address the raised concerns. Namely, I appreciate the discussion of representational alignment, the clarification of the decoding strategy, and the training probe training data size. I acknowledge the gap to model retraining-based approaches, and agree they may not need to be included.
> > I still remain skeptical of the justification/expressiveness of the two-parameter "generalized alignment" method, as well as the low-rank intervention. Although they seem to be rather empirically useful, at least a qualitative justification for the next revision would be highly appreciated.
> > Still, for the provided efforts, I am happy to raise my score.

---

> > > ### Author Response · Authors · 2026-04-07
> > >
> > > We thank the reviewer for the final feedback.
> > >
> > > We agree that the term “generalized” may sound like a broader theoretical framework than our current two-parameter formulation warrants. That was not our intent, and we will rename it. To clarify, our intent was to add a control to adjust the sparsity of prediction confidence along two complementary axes: (1) among MCQ options ($\alpha$), and (2) within each option ($\beta$). Specifically, increasing $\alpha$ amplifies the knowledge logits, in which alpha serves as inverse temperature. Consequently, the resulting probability distribution of predictions over MCQ options becomes more sparse, and more confident knowledge is more heavily weighted. On the other hand, increasing $\beta$ pushes logit values to either extreme (positive or negative), which effectively makes initially positive options more confidently positive and initially negative options more confidently negative. In this sense, it could be interpreted as making the knowledge signal stronger and "sharpening" model prediction. Although we originally named this two-parameter alignment "generalized alignment" due to its effects on both inter-option and intra-option levels, we will rename this section to “Extended Alignment” in the final version.
> > >
> > > Regarding the low-rank intervention, we acknowledge that restricting the intervention to a low-rank subspace can impact alignment accuracy. By using non-linear probes (e.g., multilayer perceptrons), we could find richer and more accurate feature spaces. The alignment between the two spaces in this case, however, would not be as straightforward, simple, and fast as the linear transformation in KAPPA, which is why we opted for linear probes as a first attempt. That said, we view this as a promising and logical direction for future work and will explicitly discuss the limitations and implications of linear subspaces in the revision.

---

### Official Review · Reviewer_HvvC · 2026-03-08

**Soundness:** 3
**Presentation:** 3
**Significance:** 3
**Originality:** 3
**Overall Recommendation:** 5
**Confidence:** 4

**Summary:**

This work is aimed at addressing the gap between the knowledge an LLM possess and its ability to pick the correct option amongst various multiple choice options presented. The authors term this gap as the “Knowledge-prediction gap.” This gap is demonstrated through an exploration of the residual streams of models using linear probes. Specifically, a linear probe is trained to predict the correct output label based on the residual stream and simultaneously a different probe is trained to predict the final prediction made by the model. The difference in performance of these probes is presented as a way to explore the knowledge prediction gap.

The authors then perform a geometric analysis of these two spaces. Specifically, the weights of the two probes are interpreted as axes for knowledge and prediction within the residual stream. The work demonstrates that those tasks where there is a large gap also demonstrate a large misalignment (correlation).

Finally, the authors present a mechanism for bridging this gap. Specifically, they present KAPPA (knowledge -alignment prediction through projection based adjustment) which reduces this misalignment. KAPPA aligns the prediction subspace with the knowledge subspace at each decoding step. This provides moderate improvements on tasks, with the highest improvement being where the knowledge prediction gap is largest.

**Compliance With Llm Reviewing Policy:**

Affirmed.

**Final Justification:**

The rebuttal addressed my main concerns and I have changed my evaluation. Some parts of my initial assessment have been reinforced. The final score reflects my updated evaluation of the work.

**Key Questions For Authors:**

There seems to be an implied causality claim. Why must the prediction capability space be aligned with the knowledge space? If these represent how “well” the model does on those axes, then why does aligning them improve performance?

Isn’t the prediction capability common across all tasks? Can you identify a common subspace across tasks?

I see two sources of the knowledge-prediction gap to be scale and distillation. The distillation process is likely to have provided smaller models with the appropriate “knowledge”, but the prediction capability itself might require larger scale models. What do your experiments across models suggest in this regard?

**Limitations:**

Yes

**Strengths And Weaknesses:**

Soundness:
Overall the methods presented are sound. The geometric analysis isn’t necessarily causal and is closer to suggestive.

The experimental setup is careful. However, the experimental setup is limited to relatively “small” models. The majority of experiments are limited to small/mid scale models. Given that larger models are more powerful, it is reasonable to assume that a larger model would be better at prediction. I understand the difficulty of evaluating large models, but I’m afraid the question is pertinent to the results presented.

The fact that different tasks have different prediction subspaces (section 5) is interesting. If that is the case, it is not clear that the prob/geometry is extracting the subspace relevant to prediction. That is, unless the subspace is rotated. A comparison of the prediction subspace across tasks would provide valuable insights into how specific that is to a task.


Presentation:
This paper is well written. The narrative is clear as is the description of the motivation and methods. The results are easy to interpret.


Significance:
The ability to pin-point the knowledge-performance gap in the residual stream through probing is significant. However, it is not clear how general this is (scale/non-distilled models). Regardless, as an initial exploration of this phenomenon this work is significant to some extent.


Originality: This work combines two different streams of work and is therefore original. Specifically, prior work either explores the knowledge-prediction gap or provides mechanisms to intervene. The combination is original.

---

> ### Author Rebuttal · Authors · 2026-03-31
>
> We thank the reviewer for recognizing the significance of our findings and initial exploration, as well as the soundness and originality of our work.
>
> ## W1. Limited evaluation on large-scale models
> We agree that understanding how the knowledge-prediction gap varies with model scale is important. Below, we present an extended analysis using a larger-scale model (Qwen3-32B).
>
> https://anonymous.4open.science/r/KAPPA-Rebuttal-66D2/R2-1.pdf
>
> On the TruthfulQA benchmark, we find that although base prediction accuracy improves with scale, **the gap between base prediction and knowledge probe accuracy remains at 9.6%. KAPPA consistently improves performance and reduces the gap across all model sizes**, suggesting that underutilization of internal knowledge persists in larger models. We will incorporate these results in the revision.
>
> ## W2. Lack of analysis on cross-task consistency of prediction subspaces
> To better understand the task specificity of the learned subspaces, we perform a geometric analysis across task pairs, comparing prediction subspaces with each other and knowledge subspaces with each other using principal angles and orthogonal linear CKA.
>
> https://anonymous.4open.science/r/KAPPA-Rebuttal-66D2/R2-2.pdf
>
> **Prediction subspaces are largely task-specific**: pairwise angles are generally large (Llama 3.1 8B: 65.4°–83.8°; Qwen2.5 7B: 79.3°–85.3°), and CKA is near-zero (typically 0.00–0.04). Knowledge subspaces show similar or stronger separation (angles up to 87°, near-zero CKA). This indicates that both are not simple rotations of a shared subspace.
>
> **The structure is not arbitrary**: a few related tasks exhibit modest alignment (e.g., BBH-Algorithmic vs. BBH-NLP prediction subspaces in Llama: 67.9°, CKA 0.15–0.23; BBQ-Religion vs. BBQ-Age knowledge subspaces in Qwen: 71.7°, CKA 0.16), suggesting limited shared structure for semantically related tasks.
>
> Within-task layerwise analyses further show that **the prediction subspace aligns with the output unembedding space $W_U$ in late layers**, indicating that it captures output-facing prediction structure rather than an arbitrary direction.
>
> Overall, these findings suggest that the learned subspaces are largely task-specific rather than rotated versions of a shared subspace, although limited structural similarity emerges for certain related tasks. We will include this analysis in the appendix of the revision.
>
> ## Q1. Justification for aligning knowledge and prediction spaces
> > Why must the prediction capability space be aligned with the knowledge space?
>
> First, we clarify that the axes do not represent “capabilities.” The knowledge probe reflects **the model’s internal confidence in each option being correct**, while the prediction probe reflects its **confidence in the chosen output.** Aligning the two thus aligns output confidence with internal correctness confidence.
>
> Second, we identify cases where **correct-answer information is linearly accessible but underutilized** in the model’s prediction. We can think of KAPPA as improving the model's generation "capability" to better leverage this underexploited knowledge.
> > Why does aligning them improve performance?
>
> Our method is not explicitly designed to optimize task performance. The observed performance gains arise as a **natural consequence of adjusting the model’s predictions toward the knowledge signal**, which is often more reliable and accurate than its final output.
> > There seems to be an implied causality claim.
>
> While we do not assume causality a priori, **our intervention experiments provide causal evidence**: applying KAPPA consistently increases AGR, decreases KLD, and often improves ACC.
>
> ## Q2. The generality of the prediction subspace across tasks
> See response to W2.
>
> ## Q3. Role of scale and distillation in the knowledge-prediction gap
>
> Our results, discussed in the W1 response, provide partial evidence disentangling the roles of distillation and scale.
>
> First, comparing Qwen3 4B and 32B, the gap in knowledge probe accuracy is relatively small (\~10%), while the gap in the accuracy of the base models is much larger (\~23%). This suggests that **distillation transfers a substantial portion of knowledge, but smaller models struggle to translate it into correct predictions.**
>
> Second, the gap is not solely driven by distillation. Non-distilled models such as Mistral v0.3 7B and Llama-3.1 8B also exhibit substantial gaps (28.8% and 19.6%), indicating that the **phenomenon persists independently of distillation.**
>
> Third, consistent with the reviewer’s intuition, **the gap generally decreases with scale, but does not disappear** (9.6% for Qwen3 32B), suggesting that scaling alone is insufficient to resolve the misalignment.
>
> Overall, these findings indicate that the knowledge-prediction gap arises from **multiple factors, including scale and training dynamics** (including but not limited to distillation).

---

> > ### Author Rebuttal · Reviewer_HvvC · 2026-04-03
> >
> > Thank you for the response and the additional analysis.
> >
> > Please consider including the "multiple factors, including scale and training dynamics" that affect the gap more explicitly in a final version of your paper. I found this very interesting.
> >
> > I am happy to increase my score by one point.

---

### Official Review · Reviewer_ecLd · 2026-03-12

**Soundness:** 3
**Presentation:** 3
**Significance:** 2
**Originality:** 3
**Overall Recommendation:** 4
**Confidence:** 3

**Summary:**

This paper investigates the knowledge–prediction gap in LLMs on multiple-choice questions (MCQs), and confirms that this gap is widespread across different models and MCQ benchmarks, with especially pronounced effects on reasoning, truthfulness, and bias-related tasks. Through a geometric analysis of the model residual stream, the paper shows that the gap fundamentally arises from a systematic misalignment between the knowledge subspace and the prediction subspace. It then proposes KAPPA, a lightweight inference-time intervention method that aligns these two subspaces via an affine transformation. Experiments on five representative model families, including Llama 3.1 8B and Qwen 2.5 7B, as well as multiple MCQ benchmarks, show that KAPPA effectively reduces the knowledge–prediction gap, improving accuracy by up to 29.3 percentage points. The method also generalizes to free-form generation settings. In addition, the study reveals a degree of transferability of these subspaces across tasks of the same type, offering a new representation-level alignment perspective for improving the faithfulness of LLM predictions.

**Compliance With Llm Reviewing Policy:**

Affirmed.

**Final Justification:**

After considering the comments from other reviewers, I decided to raise my score.

**Key Questions For Authors:**

See Weaknesses.

**Limitations:**

yes

**Strengths And Weaknesses:**

Strength:

- The paper offers a promising perspective on whether model failures arise from a lack of knowledge or from a failure to use knowledge already encoded in the model’s internal representations.

- KAPPA is simple, lightweight in formulation, and operationally clear.

- The authors conduct extensive experiments, including cross-dataset transfer, free-form generation, and broader capability assessment. Overall, the empirical study is understandable and reasonably thorough.



Weakness:

- The claim that the gap has a geometric explanation via “misaligned subspaces” is much less convincingly established. The paper does not quantify subspace relations via principal angles, CKA, Procrustes error, CCA, or even a basis-invariant distance between spans. As a result, the geometric claim currently feels more suggestive than rigorously demonstrated.

- The paper suggests that the method may transfer to free-form settings , but the current setup in Section 5.2 appears closer to a zero-shot evaluation than to realistic free-form usage. The paper also does not test how KAPPA interacts with common prompting strategies such as chain-of-thought or few-shot in-context learning. Therefore, the evidence is still insufficient to assess whether the proposed intervention remains effective in realistic generation settings. More specifically, the claim about free-form generalization seems only weakly supported by part of the datasets in Table 4.
- The empirical results suggest that the practical impact of the method is substantially stronger in some domains than in others, which weakens the overall significance claim. In particular, the largest and most actionable gains appear in reasoning and truthfulness/bias settings, whereas on more knowledge-heavy benchmarks the gap appears smaller and the intervention often yields only modest improvements.
- There is conversion of BBH and TruthfulQA into 4-choice MCQs by randomly sampling distractors. However, the paper does not report any sensitivity analysis over distractor sampling. It is therefore unclear how much of the reported gap reflects the original task, as opposed to the particular constructed MCQ version used in the experiments.

---

> ### Author Rebuttal · Authors · 2026-03-31
>
> We thank the reviewer for the encouraging feedback. We appreciate the recognition of our motivation as a promising perspective, and that our empirical study is extensive and reasonably thorough.
>
> ## W1. Insufficient geometric validation of the claim
> We agree that the geometric claim was insufficiently supported and appreciate the suggestion to quantify subspace relations more rigorously.
>
> https://anonymous.4open.science/r/KAPPA-Rebuttal-66D2/R1-1.pdf
>
> To address this, we compute **principal angles and orthogonal linear CKA** between the knowledge and prediction subspaces and their representation at each layer on TruthfulQA, GSM8K, and BBH-Algorithm, across both Llama 3.1 8B Instruct and Qwen 2.5 7B Instruct. Three findings emerge consistently:
> - (1) In the late layers selected by KAPPA, the mean principal angle between knowledge and prediction subspaces rises sharply to roughly $84^\circ–89^\circ$, indicating near-orthogonality precisely where the model forms its final answer.
> - (2) In the same layers, the angle between the prediction subspace and the unembedding matrix $W_U$ decreases, suggesting that late-layer prediction becomes increasingly output-facing while separating from the knowledge subspace.
> - (3) CKA values remain in an intermediate range (0.4–0.8), indicating that while projected representations retain some example-level similarity, the underlying spans remain far from aligned (since principal angles are near orthogonal).
>
> We further test whether this geometric separation tracks the magnitude of the behavioral gap. For Llama 3.1 8B, **the mean principal angle is strongly associated with the AGR-defined gap across eight benchmarks (Spearman $\rho = 0.976, p < 0.001$)**. For Qwen 2.5 7B, the same positive trend is observed, but does not reach significance (Spearman $\rho = 0.619, p = 0.121$).
>
> Together, these results provide **quantitative geometric evidence** that the knowledge and prediction subspaces are substantially misaligned in the layers where KAPPA intervenes.  Furthermore, it indicates that this misalignment is structurally tied to the model's output formation process. We will include these analyses in the revised manuscript.
>
> ## W2. Limited evidence for realistic free-form generalization
> We agree that the setup in Section 5.2 is closer to a zero-shot evaluation and does not fully capture more diverse interaction settings. Our goal is to provide preliminary evidence that the knowledge-prediction gap and **KAPPA extend beyond MCQ to open-ended, multi-token generation**. In this setting, KAPPA operates at each step by projecting hidden states onto the knowledge and prediction subspaces and correcting their discrepancy, thereby **influencing the entire generation trajectory**. Evaluating broader prompting strategies (e.g., CoT, few-shot ICL) is an important direction for future work.
>
> ## W3.  Uneven performance gains across domains
> **Small gains in certain domains like knowledge-heavy benchmarks are due to the LLMs having a small knowledge-prediction gap in the first place, rather than KAPPA being ineffective**. In contrast, KAPPA is effective on reasoning and truthfulness/bias tasks; this has significant implications as people are increasingly relying on LLMs for such complex and value-laden tasks (where reasoning and truthfulness/biases are critical). For reasoning tasks, the fact that KAPPA can substantially improve accuracy while bypassing long, costly reasoning trajectories is particularly valuable. More broadly, this variation in gains illustrates a contribution of our framework beyond the intervention itself: **it helps distinguish tasks where errors stem from underutilized knowledge from those where the model genuinely lacks the relevant information**.
>
> ## W4. Lack of sensitivity analysis for distractor sampling
> We thank the reviewer for highlighting this important concern, which helps strengthen the rigor of our evaluation.
>
> https://anonymous.4open.science/r/KAPPA-Rebuttal-66D2/R1-2.pdf
>
> To assess the sensitivity to distractor sampling, we repeat our experiments using 10 different random seeds for each benchmark, constructing multiple MCQ variants with independently sampled distractors. For each variant, we apply KAPPA and compute accuracy. We then evaluate whether KAPPA consistently outperforms the base model using a one-sided t-test.
> Across most settings, **KAPPA achieves statistically significant improvements over the base model**, indicating that our findings are robust to variations in distractor sampling. We report, for both the base model and KAPPA, the mean and standard deviation of accuracy across seeds, along with the corresponding p-values. We will incorporate this analysis into the revised manuscript.

---

> > ### Author Rebuttal · Reviewer_ecLd · 2026-04-03
> >
> > KAPPA is a disguised logit adjustment method, not a novel subspace alignment method. The closed-form solutions in equations (3)–(4) are equivalent to replacing the model logit with the knowledge probe logit within the predictor subspace. The contribution of the geometric framework is exaggerated and the theoretical analysis is insufficient.
> > After considering the comments from other reviewers, I decided to raise my score.

---

> > > ### Author Response · Authors · 2026-04-07
> > >
> > > We thank the reviewer for sharing their perspective. We acknowledge that, based on the current manuscript, our alignment method could be interpreted as directly aligning the model’s output logits to the knowledge probe. We believe the premise of this interpretation is the assumption that the prediction subspace is identical to the model’s output unembedding space. In what follows, we aim to (1) clarify that the two spaces are sufficiently distinct and (2) present theoretical grounding for the prediction subspace and its geometric interpretation.
> > >
> > > ## (1) Evidence for the distinction between the prediction subspace and the unembedding space
> > > As per the reviewer’s suggestion, we conducted geometric analyses by measuring principal angles and orthogonal linear CKA between the knowledge and prediction subspaces, as well as between the prediction subspace and logit space (W1 in our rebuttal). One core finding is that, at the layers where KAPPA intervenes, the **principal angle between the prediction subspace and the unembedding matrix $W_U$ remains large** (roughly $70^\circ–88^\circ$), which is far from full alignment.
> > >
> > > Moreover, if the prediction subspace simply mirrored the unembedding space and thus directly controlled the logits of the MCQ option symbols, applying KAPPA should always cause the model to generate an option symbol. However, our experiments in Section 5.2 demonstrate that the subspaces learned in an MCQ setting transfer to a free-form generation setting (where MCQ options are not provided), leading to accuracy improvements. Furthermore, KAPPA generalizes even when the target task, the answer symbols, or the number of options differ between probe training and evaluation. For example, in Section 5.1, a probe trained on TruthfulQA (option symbols: 1/2/3/4) transfers to BBQ Age (option symbols: A/B/C) with a +5.72 AGR gain. These patterns would be **difficult to explain if KAPPA simply operated as a token-level logit replacement.**
> > >
> > > ## (2) Theoretical grounding
> > > We speculate that the prediction subspace (particularly in middle layers) captures broader abstract, high-level features related to model output behavior than mere token-level logits. The KAPPA-aligned hidden states at these layers may be subsequently transformed by later layers to generate either specific answer symbols (MCQ) or corresponding response text (free-form generation) **aligned with the knowledge probe**, providing a plausible explanation for the observed transfer across output formats.
> > >
> > > This hypothesis is grounded in and consistent with prior mechanistic interpretability and activation steering literature. Specifically, the linear representation hypothesis [1] posits that residual stream activations encode high-level semantic features as different directions (while concrete token predictions are formed primarily in later layers [2, 3, 4]). Accordingly, middle-layer residual stream activations have been found to encode abstract attributes—such as emotion, personality, and values—as distinct directions [5, 6, 7], and steering the hidden states toward a specific direction consistently elicits the corresponding trait (e.g., generating "negative" responses or "hedonistic" ones). Importantly, while these directions are certainly related to the unembedding space (as they steer model outputs toward a certain trait), the two spaces are still distinct; hence, such steering does not cause the model’s outputs to converge to a single response, but rather to generate **diverse responses consistent with the target trait**. Through this lens, in KAPPA,
> > > $\tilde{W}\_{\text{pred}}\left(\tilde{W}\_{\text{pred}}^{\top}\tilde{W}\_{\text{pred}}\right)^{-1} \left(\tilde{W}\_{\text{know}}^{\top}\tilde{h}-\tilde{W}\_{\text{pred}}^{\top}\tilde{h}\right)$
> > > in the closed-form solution (Section 4.1) could be viewed as a “knowledge-aligned direction” added to hidden states, which elicits the “knowledge-alignment” behavior in model outputs in various forms.
> > >
> > > We will ensure that these geometric analyses and theoretical connections are fully incorporated into the final manuscript. We would also sincerely appreciate any suggestions for additional methods or experiments to further strengthen our theoretical analysis, and we would be happy to include them.
> > >
> > > **References:**
> > > - [1] Park, Kiho, et al. "The linear representation hypothesis and the geometry of large language models." (2023).
> > > - [2] nostalgebraist. "Interpreting GPT: the logit lens." (2020).
> > > - [3] Lad, Vedang, et al. "The remarkable robustness of LLMs: Stages of inference?" (2024).
> > > - [4] Geva, Mor, et al. "Dissecting recall of factual associations in auto-regressive language models." (2023).
> > > - [5] Tigges, Curt, et al. "Linear representations of sentiment in large language models." (2023).
> > > - [6] Jha, Shrey, et al. "Steering LLM Interactions Using Persona Vectors." (2026).
> > > - [7] Han, Jongwook, et al. "Dual Mechanisms of Value Expression: Decomposing Intrinsic and Prompted Values in Language Models." (2025).

---

### Decision · Program_Chairs · 2026-04-30

**Decision:**

Accept (regular)

**Comment:**

The paper addresses an important problem: LLMs often contain the correct answer internally but fail to predict it. The proposed KAPPA method is simple, lightweight, and consistently improves performance across multiple models and benchmarks. Reviewers found the paper well written, empirically strong, and novel in combining probing, geometric analysis, and inference-time intervention. While the geometric interpretation is somewhat suggestive and the evaluation is mostly limited to MCQ tasks and smaller models, these concerns do not outweigh the contribution.